# Towards an Explainable Comparison and Alignment of Feature Embeddings

**Mohammad Jalali** [1]   **Bahar Dibaei Nia** [2]   **Farzan Farnia** [1]

## Abstract

While several feature embedding models have been developed in the literature, comparisons of these embeddings have largely focused on their numerical performance in classification-related downstream applications. However, an interpretable comparison of different embeddings requires identifying and analyzing mismatches between sample groups clustered within the embedding spaces. In this work, we propose the *Spectral Pairwise Embedding Comparison (SPEC)* framework to compare embeddings and identify their differences in clustering a reference dataset. Our approach examines the kernel matrices derived from two embeddings and leverages the eigendecomposition of the kernel difference matrix to detect sample clusters that are captured differently by the two embeddings. We present a scalable implementation of this kernel-based approach, with computational complexity that grows linearly with the sample size. Furthermore, we introduce an optimization problem using this framework to align two embeddings, ensuring that clusters identified in one embedding are also captured in the other model. We provide numerical results demonstrating the SPEC's application to compare and align embeddings on large-scale datasets such as ImageNet and MS-COCO. The project page is available at https://mjalali.github.io/SPEC/.

## 1. Introduction

Several mainstream frameworks in computer vision and natural language processing rely on embedding models to map raw image and text inputs into spaces with semantically meaningful features (Radford et al., 2021; Liu et al., 2023; Alayrac et al., 2022; Li et al., 2023; Oquab et al., 2024). The application of pre-trained embeddings has enabled scalable solutions for many downstream tasks, particularly in scenarios where the available sample size and compute resources are significantly limited. In such cases, the embedded data can be used to train simple models, such as linear or k-nearest neighbors (KNN) classifiers, to achieve satisfactory results. Additionally, features extracted by standard embedding models are widely employed for the automated evaluation of generative models (Heusel et al., 2017; Hessel et al., 2021; Stein et al., 2023; Ospanov et al., 2025), providing accurate rankings of generative modeling architectures without requiring time-intensive human assessments.

While recent advancements in the machine learning community have introduced various embedding models that achieve remarkable results on standard image, text, and video domains, comparisons of these embeddings have primarily focused on evaluating their performance in standard downstream tasks, such as classification accuracy on benchmark datasets (e.g. ImageNet). However, such comparisons often lack interpretability and do not reveal how differently the embeddings behave in recognizing various sample types. A more fine-grained comparison is necessary to disclose explainable differences between embedding models, particularly in identifying which samples are clustered differently according to the models. Understanding these differences can aid in interpreting and debugging embeddings and can also be leveraged to align multiple embeddings. Furthermore, interpreting the discrepancies of embeddings can be used to select representation models for downstream tasks.

In this work, we propose a spectral approach called *Spectral Pairwise Embedding Comparison (SPEC)* for the fine-grained comparison of two embeddings. The SPEC framework detects differences in sample clusters assigned by two embeddings, identifying major data groups that are clustered differently by one embedding compared to the other. To achieve this, we extend the well-established kernel principal component analysis (Kernel-PCA) approach (Schölkopf et al., 1998), that leverages the eigendecomposition of the kernel similarity matrix, and propose analyzing the principal eigenvectors of the difference of embeddings' kernel matrices to interpret the differences in cluster assignments. Our analysis suggests that the SPEC framework could effectively detect cluster differences between two embeddings,

---

[1]The Chinese University of Hong Kong [2]Sharif University of Technology. Correspondence to: Mohammad Jalali <mjalali24@cse.cuhk.edu.hk>, Bahar Dibaei Nia <bahar.dibaeinia@sharif.edu>, Farzan Farnia <farnia@cse.cuhk.edu.hk>.

*Proceedings of the 42$^{nd}$ International Conference on Machine Learning*, Vancouver, Canada. PMLR 267, 2025. Copyright 2025 by the author(s).

## An overview of SPEC method

*Figure 1.* Overview of the Spectral Pairwise Embedding Comparison (SPEC) framework: The SPEC performs an eigendecomposition of the difference of kernel matrices following the two compared embeddings (e.g., DINOv2 and CLIP image embeddings) on a given reference dataset. Every eigenvector can be interpreted as a differently captured sample cluster by the embeddings, and the corresponding eigenvalue quantifies the difference between the cluster frequencies in the embedding spaces.

particularly when the clusters can be separated using the principal eigendirections of the kernel matrices.

To address the computational challenges of performing eigendecomposition on large-scale datasets, we develop a scalable implementation of SPEC. A direct eigendecomposition of the $n \times n$ kernel difference matrix requires $O(n^3)$ computations for a dataset with $n$ samples, which is computationally expensive for large datasets. Assuming a bounded feature dimension $d$ for the applied kernel function, we prove that the eigenspace of the kernel difference matrix can be computed using $O(d^2n + d^3)$ operations, resulting in a scalable algorithm under a moderate dimension $d$ value. Furthermore, we extend this scalable computation method to shift-invariant kernel functions, e.g. the Gaussian (RBF) kernel, by employing the framework of random Fourier features (RFF) (Rahimi & Recht, 2007), where the size of RFF proxy-feature map can be controlled for a more efficient application of SPEC.

We also explore the application of the SPEC framework to define a distance measure between two embeddings. We define the SPEC-diff distance as the spectral radius of the kernel difference matrix, which aims to quantify the weight of the most differently captured cluster in one embedding that is not strongly clustered by the other model. We dis-

cuss scalable computations of this distance and its gradient with respect to the embedding parameters. Using the power method and the calculated left and right eigenvectors of the differential covariance matrix, we enable gradient-based optimization of the distance measure for aligning embedding models. This gradient-based approach leads to a method we call *SPEC-align*, aligning embedding models by minimizing their differences in clustering a reference dataset. SPEC-align is particularly useful for aligning cross-modality embeddings, such as CLIP (Radford et al., 2021), with a state-of-the-art single-modality embedding. The alignment can improve the performance of cross-modality embeddings in capturing concepts specific to individual modalities.

Finally, we present numerical experiments on several standard image and text embeddings using benchmark datasets. Our results demonstrate the scalability of the SPEC framework in revealing differences in sample clusters across embeddings over large-scale datasets. In our experiments, we tested the SPEC algorithm's application with both cosine similarity and shift-invariant Gaussian kernels, where we leverage random Fourier features for the latter case. Additionally, we discuss the application of SPEC-align to align the CLIP model with single-modality embeddings. The empirical results highlight the effectiveness of SPEC-align in reducing the differences between CLIP's image embeddings

and specialized image-domain embeddings. The following is a summary of our work's main contributions:

- Proposing the SPEC framework for explainable comparison of two embeddings,

- Providing a scalable SPEC implementation with linearly growing computational cost to the sample size,

- Developing the gradient-based SPEC-align method to align two embeddings and matching their sample clusters,

- Demonstrating the successful application of SPEC in comparing and aligning embeddings on benchmark datasets.

## 2. Related Work

**Spectral Clustering, Kernel PCA, and Random Fourier Features**. Kernel PCA (Schölkopf et al., 1998) is a widely established method for dimensionality reduction that relies on the eigendecomposition of the kernel matrix. Several studies (Bengio et al., 2003b;a) have explored the relationship between kernel PCA and spectral clustering. Also, the analysis of random Fourier features (Rahimi & Recht, 2007) for performing scalable kernel PCA has been studied by Chitta et al. (2012); Ghashami et al. (2016); Ullah et al. (2018); Sriperumbudur & Sterge (2022); Gedon et al. (2023). In our work, we introduce a spectral approach for comparing two embeddings, leveraging the random Fourier features framework to address computational challenges.

**Evaluation and Comparison of Embeddings.** Embedding evaluation is typically conducted using a limited set of downstream tasks (Chen et al., 2013; Santos et al., 2020; Perone et al., 2018; Choi et al., 2021). Existing NLP benchmarks (Gao et al., 2021; Reimers & Gurevych, 2019) focus on limited tasks. Muennighoff et al. (2023) introduces MTEB, standardizing text embedder evaluation across diverse NLP tasks. In Image embeddings, Kynkäänniemi et al. (2023); Stein et al. (2023) compared different image embeddings and showed how they can influence different tasks, specifically the evaluation of generative models. Another line of research is probing methods (Belinkov, 2022; Pimentel et al., 2020; Adi et al., 2017; Rogers et al., 2021), which analyze model embeddings by training small models on them to understand what information is encoded. These methods help assess how well embeddings capture specific features, although they are not focused on embedding comparison.

Darrin et al. (2024) propose a new metric for comparing embeddings without labeled data and propose the concept of information sufficiency (IS) to quantify the required information to simulate one embedding from another. Our work offers a complementary, explainable method for comparing embeddings by detecting different sample clusters assigned by embeddings and providing a method for aligning them.

Also, Vargas et al. (2024) discuss a spectral approach to the comparison of embeddings, where they consider the PCA-principal components of the collective embedded vectors of two representations. On the other hand, our work considers the kernel difference matrix, which better suits the comparison of embeddings in different dimensions and leads to a distance metric for their alignment.

A different yet related line of work is the evaluation of generative models. (Bińkowski et al., 2018; Jalali et al., 2023; Ospanov et al., 2024; Jalali et al., 2024; Ospanov & Farnia, 2025; Wang et al., 2023b; Hu et al., 2025a; Rezaei et al., 2025; Hu et al., 2025b;c) leverage the eigenspectrum of kernel matrices to quantify diversity. The papers (Jiralerspong et al., 2023; Zhang et al., 2024) explore novelty evaluation, analyzing how generated samples differ from those of a reference model. In particular, (Zhang et al., 2024; 2025) propose a spectral method for measuring the entropy of the novel modes of a generative model with respect to a reference model.

**Embeddings Alignment.** There are many works on embedding alignment for multimodal models (Bellagente et al., 2023; Lu et al., 2024; Han et al., 2024; Wang et al., 2023c; Girdhar et al., 2023; Grave et al., 2019). Salman et al. (2024); Eslami & de Melo (2025) introduced a method, which demonstrates that adversarial perturbations can force text embeddings to align with any image in multimodal models, exposing security vulnerabilities in vision-language learning. Ye et al. (2024) proposed ModalChorus, an interactive system that visualizes and corrects misalignments in multi-modal embeddings, improving interpretability and optimization. Focusing on fine-grained alignment, Yin et al. (2024) introduced a method for explicitly aligning individual word embeddings with corresponding visual features, leveraging cross-modal attention to refine token-image associations. In contrast, our work focuses on aligning embeddings in a kernel setting to match their sample clusters.

## 3. Preliminaries

### 3.1. Embedding maps and spaces

Consider a data vector $x \in \mathcal{X}$ in the space $\mathcal{X}$. An embedding map $\psi : \mathcal{X} \rightarrow \mathcal{S}$ maps an input $x$ to the embedding space $\mathcal{S}$, which is supposed to provide a more meaningful representation of the input data vector. Throughout this work, we focus on the problem of characterizing and interpreting the differences of two embedding maps $\psi_1 : \mathcal{X} \rightarrow \mathcal{S}_1$ and $\psi_2 : \mathcal{X} \rightarrow \mathcal{S}_2$, which can map the input $x \in \mathcal{X}$ to different embedding spaces $\mathcal{S}_1, \mathcal{S}_2$.

### 3.2. Kernel Functions and Covariance Matrix

A kernel function $k : \mathcal{X} \times \mathcal{X} \rightarrow \mathbb{R}$ maps two inputs $x, x'$ to a similarity score $k(x, x') = \langle \phi(x), \phi(x') \rangle \in [0, 1]$ that is the

inner product of the representation of $x, x'$ characterized by $\phi : \mathcal{X} \to \mathbb{R}^d$. This definition implies that for every sequence of data points $x_1, \ldots, x_n \in \mathcal{X}$, the following kernel matrix is positive semi-definite (PSD):

$$K = \begin{bmatrix} k(x_1, x_1) & \cdots & k(x_1, x_n) \\ \vdots & \ddots & \vdots \\ k(x_n, x_1) & \cdots & k(x_n, x_n) \end{bmatrix} \succeq \mathbf{0} \qquad (1)$$

Well-known examples of kernel functions include the cosine-similarity kernel $k_{\text{cosine}}(x, y) = \frac{x^\top y}{\|x\|_2 \|y\|_2}$ and the Gaussian (RBF) kernel defined for bandwidth $\sigma$ as

$$k_{\text{Gaussian}(\sigma^2)}(x, y) = \exp\Big( \frac{-\|x - y\|_2^2}{2\sigma^2} \Big)$$

Both these examples are normalized kernels where $k(x, x) = 1$ holds for every $x \in \mathcal{X}$. Note that the kernel matrix in (1) can be written as $K = \Phi \Phi^\top$ where $\Phi \in \mathbb{R}^{n \times d}$ contains $\phi(x_i)$ as its $i$th row for $i \in \{1, \ldots, n\}$. Then, the kernel covariance matrix $C_X \in \mathbb{R}^{d \times d}$ can be defined by reversing the matrix multiplication order as:

$$C_X := \frac{1}{n} \Phi^\top \Phi = \frac{1}{n} \sum_{i=1}^{n} \phi(x_i) \phi(x_i)^\top \qquad (2)$$

Therefore, $C_X = \frac{1}{n} \Phi^\top \Phi$ and $\frac{1}{n} K = \frac{1}{n} \Phi \Phi^\top$ share the same non-zero eigenvalues, since they represent the products of matrices with flipped multiplication orders.

# 4. SPEC: A Spectral Identification of Embeddings' Mismatches

Consider a set of $n$ data points $x_1, \ldots, x_n \in \mathcal{X}$ and two embedding maps $\psi_1 : \mathcal{X} \to \mathcal{S}_1$ and $\psi_2 : \mathcal{X} \to \mathcal{S}_2$. Also, suppose $k_1 : \mathcal{S}_1 \times \mathcal{S}_1 \to \mathbb{R}$ and $k_2 : \mathcal{S}_2 \times \mathcal{S}_2 \to \mathbb{R}$ are kernel functions to be applied to the embedding spaces for $\psi_1, \psi_2$, respectively.

To compare the two embeddings, note that their corresponding spaces $\mathcal{S}_1$ and $\mathcal{S}_2$ may have different dimensions, and therefore a sample-specific comparison of the embedded vectors $\psi_1(x), \psi_2(x)$ for each individual data point $x$ will not provide a meaningful comparison of the embedding maps. Therefore, a more relevant approach is to compare the embeddings' outputs over the entire set of data $\{x_1, \ldots, x_n\}$ and investigate which structures are dissimilar between the sets of embedded data following the embeddings. Here, we consider a spectral approach and particularly focus on the difference of kernel matrices between the two embeddings. In the following, we discuss how the eigenspace of the kernel difference matrix can help identify the differently clustered points by the two embeddings. In the following, we denote the kernel matrix of the first embedding with $K_{\psi_1} = \big[ k_1(\psi_1(x_i), \psi_1(x_j)) \big]_{(i,j)=(1,1)}^{(n,n)}$ and

the kernel matrix of the second embedding with $K_{\psi_2} = \big[ k_2(\psi_2(x_i), \psi_2(x_j)) \big]_{(i,j)=(1,1)}^{(n,n)}$.

**Definition 1.** We define the normalized kernel difference matrix $\Lambda_{\psi_1, \psi_2} \in \mathbb{R}^n$ as follows:

$$\Lambda_{\psi_1, \psi_2} := \frac{1}{n} \Big( K_{\psi_1} - K_{\psi_2} \Big) \qquad (3)$$

We propose the framework of *Spectral Pairwise Embedding Comparison (SPEC)* where the two embeddings $\psi_1$ and $\psi_2$ are compared using the eigendirections of the kernel difference matrix $\Lambda_{\psi_1, \psi_2}$. As we will show, the principal eigenvectors can be interpreted as the clusters of samples assigned by embedding $\psi_1$ that are less strongly grouped by the second embedding $\psi_2$. In what follows, we first show a theoretical result supporting the mentioned property of $\Lambda_{\psi_1, \psi_2}$'s eigenvectors. Next, we provide a scalable computation method for computing the eigenspace of $\Lambda_{\psi_1, \psi_2}$ that linearly scales with the sample size $n$.

Theorem 1 proves that under the following two conditions on the sample index set $\mathcal{I} \subset \{1, \ldots, n\}$, the eigndirections of $\Lambda_{\psi_1, \psi_2}$ can separate the clustered sample indices from the rest of samples. Note that the notation $\mathcal{I}^c$ denotes the complement index set of $\mathcal{I}$, and $K[\mathcal{I}, \mathcal{J}]$ denotes the sub-matrix of $K$ with rows in $\mathcal{I}$ and columns in $\mathcal{J}$.

- **Condition 1**: Suppose the sample set $X_\mathcal{I}$ characterized by index set $\mathcal{I}$ are separated from the rest of samples by embedding $\psi_1$, where the normalized block kernel matrix $\frac{1}{n} K_{\psi_1}[\mathcal{I}, \mathcal{I}^c]$ has an $\epsilon_1$-bounded Frobenius norm:

$$\Big\| \frac{1}{n} K_{\psi_1}[\mathcal{I}, \mathcal{I}^c] \Big\|_F \le \epsilon_1$$

- **Condition 2**: Suppose the sample set $X_\mathcal{I}$ characterized by index set $\mathcal{I}$ are weakly grouped by embedding $\psi_2$, where the normalized block kernel matrix $\frac{1}{n} K_{\psi_2}[\mathcal{I}, \mathcal{I}]$ has an $\epsilon_2$-bounded $\ell_2$-operator norm (which is the maximum eigenvalue for this PSD matrix)

$$\Big\| \frac{1}{n} K_{\psi_2}[\mathcal{I}, \mathcal{I}] \Big\|_2 \le \epsilon_2$$

**Theorem 1.** *Consider the kernel difference matrix $\Lambda_{\psi_1, \psi_2}$ in (3). Suppose Conditions 1 and 2 hold. Let $\mathbf{v}_1, \ldots, \mathbf{v}_n$ be the unit-norm eigenvectors of $\Lambda_{\psi_1, \psi_2}$ corresponding to eigenvalues $\lambda_1, \ldots, \lambda_n$. For every $i \in \{1, \ldots, n\}$, we define $\lambda_i^\mathcal{I}$ and $\lambda_i^{\mathcal{I}^c}$ to be the closest eigenvalue of $\Lambda_{\psi_1, \psi_2}[\mathcal{I}, \mathcal{I}]$ and $\Lambda_{\psi_1, \psi_2}[\mathcal{I}^c, \mathcal{I}^c]$ to $\lambda_i$. Then, the following holds for $\xi = 4(\epsilon_1^2 + \epsilon_2)$:*

$$\sum_{i=1}^{n} \big( \lambda_i - \lambda_i^\mathcal{I} \big)^2 \big\| \mathbf{v}_i[\mathcal{I}] \big\|_2^2 + \big( \lambda_i - \lambda_i^{\mathcal{I}^c} \big)^2 \big\| \mathbf{v}_i[\mathcal{I}^c] \big\|_2^2 \le \xi$$

*Proof.* We defer the proof of the theoretical statements to the Appendix A.1. □

**Corollary 1.** *In the setting of Theorem 1, suppose* $\mathbf{v}$ *is an eigenvector of* $\Lambda_{\psi_1,\psi_2}$ *for eigenvalue* $\lambda$ *whose gap with the maximum eigenvalue of the sub-matrix* $\Lambda_{\psi_1,\psi_2}[\mathcal{I}^c,\mathcal{I}^c]$ *satisfies* $\lambda - \lambda_{\max}(\Lambda_{\psi_1,\psi_2}[\mathcal{I}^c,\mathcal{I}^c]) \geq \gamma > 0$. *Then,*

$$\left\|\mathbf{v}[\mathcal{I}^c]\right\|_2 \leq \frac{2\sqrt{\epsilon_1^2 + \epsilon_2}}{\gamma}.$$

The above corollary proves that if an eigenvalue $\lambda$ of the kernel difference matrix $\Lambda_{\psi_1,\psi_2}$ is sufficiently large, such that its gap with the maximum eigenvalue of the block $\Lambda_{\psi_1,\psi_2}[\mathcal{I}^c,\mathcal{I}^c]$ (with the complement of samples in $\mathcal{I}$ clustered by $\psi_1$ yet not by $\psi_2$) is higher than the threshold $\lambda$, then the $\mathcal{I}^c$-entries of the corresponding unit-norm eigenvector $\mathbf{v}$ will be bounded, reflecting the lack of $\mathcal{I}^c$ samples in the differentially clustered samples by embedding $\psi_1$ and $\psi_2$. Based on the above theoretical results, we propose considering the principal eigenvectors of the kernel difference matrix, and using their significant-value entries to find the subset of samples clustered by embedding $\psi_1$ but not grouped by $\psi_2$.

Since the kernel difference matrix is of size $n \times n$, a standard eigendecomposition will cost $O(n^3)$ computations. Proposition 1 shows that the computation cost will be lower for embeddings with bounded feature maps. In fact, this result shows the computation of the eigenspace can be performed using linearly growing computation cost $O(n)$.

**Proposition 1.** *Consider the kernel difference matrix* $\Lambda_{\psi_1,\psi_2}$ *in* (3). *This matrix shares the same non-zero eigenvalues with the following matrix:*

$$\Gamma_{\psi_1,\psi_2} = \begin{bmatrix} C_{\psi_1} & C_{\psi_1,\psi_2} \\ -C_{\psi_1,\psi_2}^\top & -C_{\psi_2} \end{bmatrix} \in \mathbb{R}^{(d_1+d_2)\times(d_1+d_2)} \quad (4)$$

*where* $C_{\psi_1} \in \mathbb{R}^{d_1 \times d_1}, C_{\psi_2} \in \mathbb{R}^{d_2 \times d_2}$ *are the kernel covariance matrices of* $\psi_1, \psi_2$, *respectively, and* $C_{\psi_1,\psi_2} \in \mathbb{R}^{d_1 \times d_2}$ *is the cross-covariance matrix, defined as:*

$$C_{\psi_1} := \frac{1}{n}\sum_{i=1}^{n} \phi_1\big(\psi_1(\mathbf{x}_i)\big)\phi_1\big(\psi_1(\mathbf{x}_i)\big)^\top,$$

$$C_{\psi_2} := \frac{1}{n}\sum_{i=1}^{n} \phi_2\big(\psi_2(\mathbf{x}_i)\big)\phi_2\big(\psi_2(\mathbf{x}_i)\big)^\top,$$

$$C_{\psi_1,\psi_2} := \frac{1}{n}\sum_{i=1}^{n} \phi_1\big(\psi_1(\mathbf{x}_i)\big)\phi_2\big(\psi_2(\mathbf{x}_i)\big)^\top.$$

We also note that for every (right) eigenvector $\mathbf{v} \in \mathbb{R}^{d_1+d_2}$ of matrix $\Gamma_{\psi_1,\psi_2}$ in (4), which we call the *differential covariance matrix*, we can find the corresponding eigenvector $\mathbf{u}$ of kernel difference matrix $\Lambda_{\psi_1,\psi_2}$ as follows:

$$\mathbf{u} = \begin{bmatrix} \phi_1(\psi_1(x_1)) & \phi_2(\psi_2(x_1)) \\ \vdots & \vdots \\ \phi_1(\psi_1(x_n)) & \phi_2(\psi_2(x_n)) \end{bmatrix} \mathbf{v}$$

Note that the computation of the matrix $\Gamma_{\psi_1,\psi_2}$ can be performed with $O\big(n(d_1+d_2)^2\big)$ computations, linearly growing in sample size, and the eigendecomposition of $\Gamma_{\psi_1,\psi_2}$ can be handled via $O\big((d_1+d_2)^3\big)$, depending on the dimensions of the kernel feature maps $\phi_1, \phi_2$, and finally the eigenvector mapping from $\Gamma_{\psi_1,\psi_2}$ to $\Lambda_{\psi_1,\psi_2}$ will be $O\big(n(d_1+d_2)^2\big)$. Therefore, the entire eigenvector computation of $\Lambda_{\psi_1,\psi_2}$ can be handled using $O\big(n(d_1+d_2)^2 + (d_1+d_2)^3\big)$ computations. The algorithm 1 contains the main steps in computing SPEC-eigenvectors using the above approach. As detailed in this algorithm, the computation of the differential kernel covariance matrix can be run over samples in a cascade, avoiding the need to store a large dataset.

**Remark 1.** Unlike the $n \times n$ kernel difference matrix $\Lambda_{\psi_1,\psi_2} \in \mathbb{R}^{n \times n}$, the differential covariance matrix $\Gamma_{\psi_1,\psi_2} \in \mathbb{R}^{(d_1+d_2)\times(d_1+d_2)}$ is not a symmetric matrix. Still, $\Gamma_{\psi_1,\psi_2}$ will only have real eigenvalues, where the non-zero eigenvalues are shared with $\Lambda_{\psi_1,\psi_2}$. In Appendix A.3, we further provide a Cholesky decomposition-based method to reduce the computation of $\Gamma_{\psi_1,\psi_2}$'s non-zero eigenvalues and eigenvectors to the eigendecomposition of a $(d_1+d_2)\times(d_1+d_2)$-dimensional *symmetric* matrix.

Applying the standard linear and cosine-similarity kernels, the kernel feature dimension will match that of the embedding, which is usually bounded by 1000 for standard image and text embeddings. In the case of shift-invariant kernels, e.g. the Gaussian (RBF) kernel, whose feature dimension is infinite, we utilize the random Fourier features (RFFs) (Rahimi & Recht, 2007; Sutherland & Schneider, 2015) to reduce the dimension of the kernel feature dimension with a proxy kernel function characterized by the RFFs. Following the RFF framework, given a kernel function $k(x,y) = \kappa(x - y)$ that is normalized i.e. $\kappa(0) = 1$, we draw a number $m$ independent Fourier features $\omega_1, \ldots, \omega_m \sim \widehat{\kappa}$ from probability density function $\widehat{\kappa}$ which denotes the Fourier transform of $\kappa$ defined as

$$\widehat{\kappa}(\omega) = \frac{1}{(2\pi)^d}\int_{\mathcal{X}} \kappa(x)\exp(-i\langle\omega,x\rangle)\mathrm{d}x$$

Then, the RFF method approximates the shift-invariant kernel $k(x,y) \approx \widehat{k}(x,y) = \big\langle\widehat{\phi}(x),\widehat{\phi}(y)\big\rangle$ where

$$\widehat{\phi}(x) = \frac{1}{\sqrt{m}}\Big[\cos(\omega_1^\top x), \sin(\omega_1^\top x), ., \cos(\omega_m^\top x), \sin(\omega_m^\top x)\Big]$$

**Theorem 2.** *Consider normalized shift-invariant kernel* $k_1(x,y) = \kappa_1(x - y)$ *and* $k_2(x',y') = \kappa_2(x' - y')$. *Then, drawing* $m$ *Fourier features* $\omega_i^{(1)} \sim \widehat{\kappa}_1$ *and* $\omega_i^{(2)} \sim \widehat{\kappa}_2$, *we form the RFF-proxy kernel functions* $\widehat{k}_1, \widehat{k}_2$. *Then, considering eigenvalues* $\widehat{\lambda}_1, \ldots, \widehat{\lambda}_n$ *and eigenvectors* $\widehat{\mathbf{v}}_1, \ldots, \widehat{\mathbf{v}}_n$ *of proxy* $\widehat{\Lambda}_{\psi_1,\psi_2}$, *for every* $\delta > 0$, *the following holds with*

**Algorithm 1** Spectral Pairwise Embedding Comparison (SPEC)

1: **Input:** Sample set $\{\mathbf{x}_1, \ldots, \mathbf{x}_n\}$, embeddings $\psi_1$ and $\psi_2$, kernel feature maps $\phi_1$ and $\phi_2$
2: Initialize $C_{\psi_1} = \mathbf{0}_{d_1 \times d_1}$, $C_{\psi_2} = \mathbf{0}_{d_2 \times d_2}$, $C_{\psi_1,\psi_2} = \mathbf{0}_{d_1 \times d_2}$
3: **for** $i \in \{1, \ldots, n\}$ **do**
4: 	Update $C_{\psi_1} \leftarrow C_{\psi_1} + \frac{1}{n}\phi_1(\psi_1(\mathbf{x}_i))\phi_1(\psi_1(\mathbf{x}_i))^\top$
5: 	Update $C_{\psi_2} \leftarrow C_{\psi_2} + \frac{1}{n}\phi_2(\psi_2(\mathbf{x}_i))\phi_2(\psi_2(\mathbf{x}_i))^\top$
6: 	Update $C_{\psi_1,\psi_2} \leftarrow C_{\psi_1,\psi_2} + \frac{1}{n}\phi_1(\psi_1(\mathbf{x}_i))\phi_2(\psi_2(\mathbf{x}_i))^\top$
7: **end for**
8: Construct $\Gamma_{\psi_1,\psi_2}$ as in Equation (4)
9: Compute eigenvalues $\lambda_{1:d_1+d_2}$ and eigenvectors $\mathbf{v}_{1:d_1+d_2}$ of non-symmetric matrix $\Gamma_{\psi_1,\psi_2}$
10: **for** $i \in \{1, \ldots, d_1 + d_2\}$ **do**
11: 	Map eigenvector $\mathbf{u}_i = \begin{bmatrix} \phi_1(\psi_1(\mathbf{X})) & \phi_2(\psi_2(\mathbf{X})) \end{bmatrix} \mathbf{v}_i$
12: **end for**
13: **Output:** Eigenvalues $\lambda_1, \ldots, \lambda_{d_1+d_2}$, eigenvectors $\mathbf{u}_1, \ldots, \mathbf{u}_{d_1+d_2}$.

*probability at least $1 - \delta$:*

$$\sum_{i=1}^{n} \left\| \Lambda_{\psi_1,\psi_2} \widehat{\mathbf{v}}_i - \widehat{\lambda}_i \widehat{\mathbf{v}}_i \right\|_2^2 \leq \frac{128 \log(3/\delta)}{m}.$$

Theorem 2 shows the eigenvectors of the RFF-proxy kernel difference $\widehat{\Lambda}_{\psi_1,\psi_2}$ provide a proxy for the eigenspace of the target kernel difference function $\Lambda_{\psi_1,\psi_2}$. Note that the differential covariance matrix of the proxy-RFF kernel, the dimension of $\widehat{\Gamma}_{\psi_1,\psi_2}$ will be $4m \times 4m$, that is finite for every shift-invariant kernel. As a result, one can apply the eigenspace correspondence in Proposition 1 to the proxy kernel function and reduce the computational complexity to $O(m^2 n + m^3)$ for $m$ RFF features and $n$ samples.

## 5. SPEC-based Quantification of Embedding Differences

As discussed earlier, the eigenspace of the kernel difference matrix $\Lambda_{\psi_1,\psi_2}$ provides information on the differently clustered samples by the two embeddings. Therefore, the SPEC approach motivates measuring the difference of two embeddings using the eigenspectrum of $\Lambda_{\psi_1,\psi_2}$. Here, we specifically focus on the spectral radius of $\Lambda_{\psi_1,\psi_2}$, i.e., its maximum absolute eigenvalue $\rho(\Lambda_{\psi_1,\psi_2}) = \max_{1 \leq i \leq n} |\lambda_i(\Lambda_{\psi_1,\psi_2})|$ ($\rho(A)$ denotes $A$'s spectral radius). Note that $\Lambda_{\psi_1,\psi_2}$ is by definition a symmetric matrix with a zero trace, and therefore its eigenvalues are all real and add up to 0. The following definition states the difference measure, which we call *SPEC-diff* score:

$$\text{SPEC-diff}(\psi_1, \psi_2) := \rho(\Lambda_{\psi_1,\psi_2}). \qquad (5)$$

Since SPEC-diff is only a function of $\Lambda_{\psi_1,\psi_2}$'s non-zero eigenvalues, Proposition 1 shows that SPEC-diff$(\psi_1, \psi_2) = \rho(\Gamma_{\psi_1,\psi_2})$ is equal to the spectral radius of the differential covariance matrix $\Gamma_{\psi_1,\psi_2}$, therefore, it is a symmetric pseudo-distance whose computation cost scales linearly with sample size $n$.

While the SPEC-diff measure can be used to quantify the mismatches of two embeddings, it can be further optimized in the training or fine-tuning of an embedding map $\psi_{1,\theta}$'s parameters $\theta$ in order to align the embedding's clusters with another reference embedding $\psi_2$. The optimization problem to be solved for such an alignment of the embeddings will be the following, which we call the *SPEC-align* problem:

$$\min_{\theta \in \Theta} \; \mathcal{L}(\psi_{1,\theta}) + \beta \cdot \text{SPEC-diff}(\psi_{1,\theta}, \psi_2) \qquad (6)$$

In the above, $\mathcal{L}(\psi_{1,\theta})$ denotes the original loss function of training embedding $\psi_{1,\theta}$ and $\beta$ denotes the coefficient of the penalty function SPEC-diff$(\psi_{1,\theta}, \psi_2)$, penalizing the mismatch with reference embedding $\psi_2$. To apply a gradient-based optimization algorithm to solve (6), one needs to efficiently compute the gradient of SPEC-diff$(\psi_{1,\theta}, \psi_2)$ with respect to parameter $\theta$. The following proposition shows that the gradient computation can be run in $O(n_B)$ over a batch size $n_B$.

**Proposition 2.** *Consider the definitions in* (3),(4),(5). *Then, assuming a unique top eigenvalue (in terms of absolute value) for $\Gamma_{\psi_{1,\theta},\psi_2}$ with the left and right eigenvectors $\mathbf{u}_{left}, \mathbf{u}_{right}$, we will have:*

$$\nabla_\theta \text{SPEC-diff}(\psi_{1,\theta}, \psi_2) = \nabla_\theta \left( \left| \mathbf{u}_{left}^\top \Gamma_{\psi_{1,\theta},\psi_2} \mathbf{u}_{right} \right| \right) \; (7)$$

Therefore, the above proposition suggests computing the top left and right eigenvector of the $(d_1 + d_2) \times (d_1 + d_2)$ kernel difference matrix, which can be computed using the power method, and subsequently to take the gradient of the scalar function $\left| \mathbf{u}_{left}^\top \Gamma_{\psi_{1,\theta},\psi_2} \mathbf{u}_{right} \right|$ which is the absolute value of the mean of the function value for each individual sample $x_1, \ldots, x_n$. This property is especially suitable for applying stochastic gradient methods.

**Remark 2.** Another viable difference function follows the $\ell_2$-norm of the eigenvalues of $\Lambda_{\psi_1,\psi_2}$ (note that SPEC-diff is the $\ell_\infty$-norm of $\Lambda_{\psi_1,\psi_2}$'s eigenvalues). Since $\Lambda_{\psi_1,\psi_2}$ is a symmetric matrix with real eigenvalues, the squared-$\ell_2$-norm of its eigenvalues is its Frobenius norm-squared:

$$\left\| \Lambda_{\psi_1,\psi_2} \right\|_F^2 = \frac{1}{n^2} \sum_{i=1}^{n} \sum_{j=1}^{n} \left( k_{\psi_1}(x_i, x_j) - k_{\psi_2}(x_i, x_j) \right)^2.$$

An advantage of the above embedding distance function is the possibility of performing stochastic optimization, where given a mini-batch of $B$ samples, we can consider the estimation $\frac{1}{B^2} \sum_{i,j=1}^{B} \left( k_{\psi_1}(x_i, x_j) - k_{\psi_2}(x_i, x_j) \right)^2$ in the alignment process. This kernel difference Frobenius-norm-based

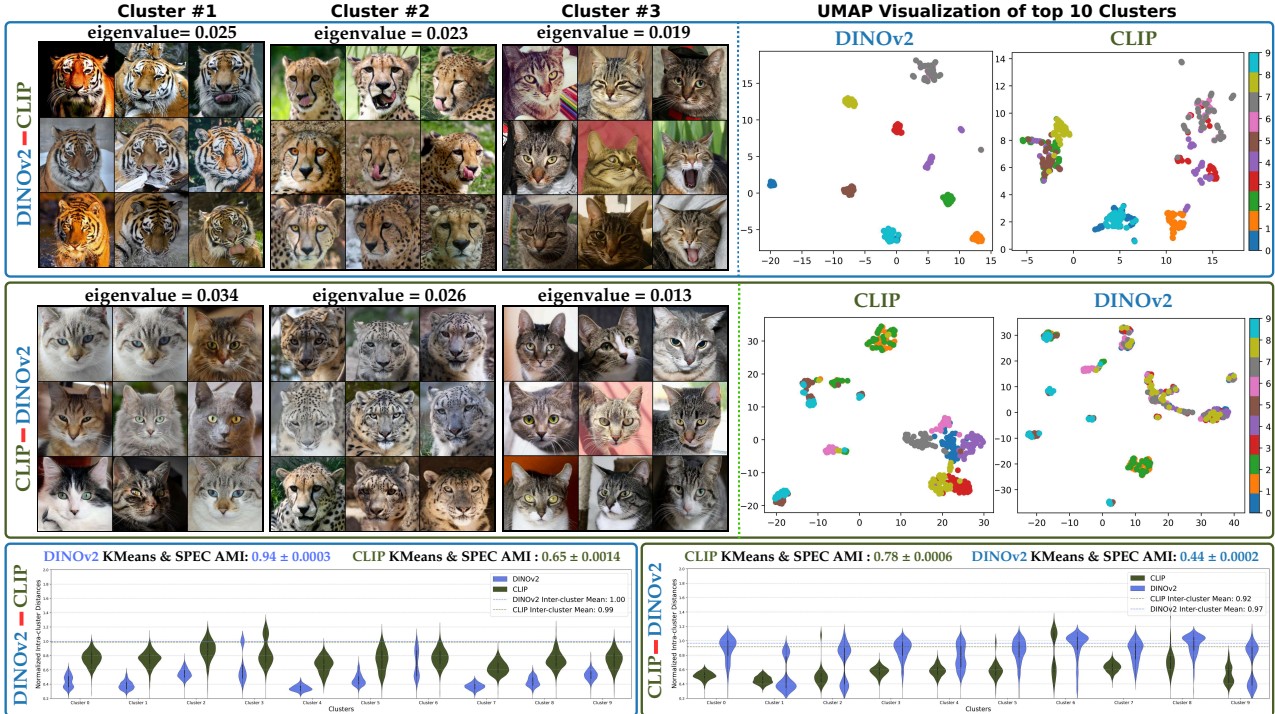

Figure 2. Comparison of different embeddings on 15K samples from the AFHQ dataset, consisting of 5K cats, 5K wildlife, and 5K dogs. The number at the top of each image represents the eigenvalue of the corresponding SPEC cluster. The last two images in each row show the UMAP representation of the SPEC clusters for each embedding individually.

alignment of embeddings has also been explored in the concurrent work by Gong et al. (2025).

# 6. Numerical Results

In this section, we first discuss the experimental settings and then apply the SPEC algorithm to compare different image and text embeddings across various large-scale datasets. Finally, we explore the use of the SPEC-align method to match the sample clusters of the embeddings.

**Datasets.** In our experiments on image data, we used four datasets: AFHQ (Choi et al., 2020) (15K animal faces in categories of cats, wildlife, and dogs), FFHQ (Karras et al., 2019) (70K human-face images), ImageNet-1K (Deng et al., 2009) (1.4 million images across 1,000 labels), and MS-COCO 2017 (Lin et al., 2015) (≈110K samples of diverse scenes with multiple objects). Additionally, similar to (Materzynska et al., 2022), we created a custom dataset derived from 10 selected classes from ImageNet-1k, where we overlaid text labels directly on images.

**Embeddings.** The feature embeddings tested in this study include the image embeddings: CLIP (Radford et al., 2021), DINOv2 (Oquab et al., 2024), Inception-V3 (Szegedy et al., 2016), and SWAV (Caron et al., 2021), and the text embeddings: RoBERTa (Liu et al., 2020), CLIP (Radford et al.,

2021), and E5-V2 (Wang et al., 2023a). All embeddings were extracted using pre-trained models, and standard preprocessing was applied for uniformity across datasets.

**Experimental settings.** In our experiments, we computed the SPEC differential kernel covariance matrix using $m = 2000$ independent random Fourier features for a Gaussian kernel. To determine the Gaussian kernel bandwidth $\sigma$, we followed the kernel-based evaluation of generative models in (Jalali et al., 2023; Ospanov et al., 2024) and selected the embeddings bandwidths such that the difference between top eigenvalue is less than 0.01. We provide the detailed SPEC algorithm in Algorithm 1. The experiments were performed on two RTX-4090 GPUs.

**SPEC comparison of different embeddings.** To evaluate SPEC, we compared various image embeddings using the AFHQ dataset. As shown in Figure 2, we employed SPEC for pairwise comparisons to analyze the difference between these embeddings. We reported the top 9 images that correspond to the maximum entries of the top three eigenvectors in the SPEC approach. Subsequently, we found and visualized the top 100 samples (with maximum entries) from each of the top 10 eigenvectors (i.e. SPEC-identified clusters). To confirm whether these samples were clustered by the first embedding and not by the second embedding, we used UMAP maps (McInnes et al., 2018) to validate

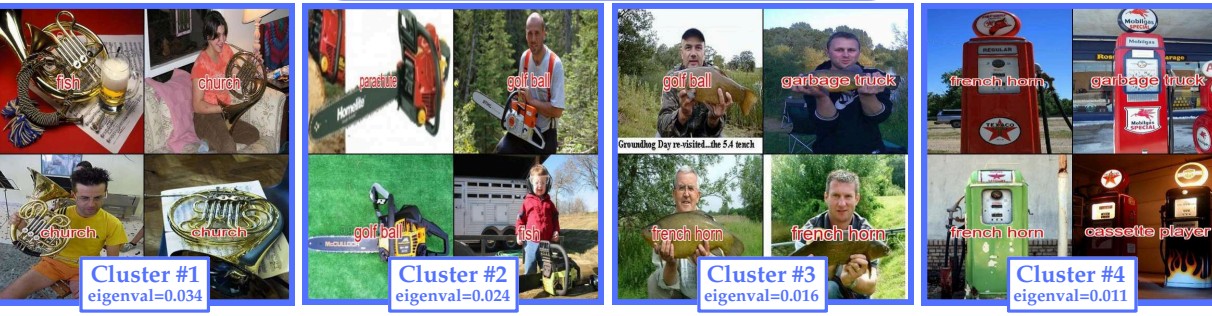

*Figure 3.* Top 4 SPEC-identified clusters comparing CLIP and DINOv2 embeddings on 10 ImageNet classes with overlaid text labels.

the SPEC-identified different-captured sample groups by the two embeddings. In the Appendix, we further provide the t-SNE (Van der Maaten & Hinton, 2008) and PaCMAP (Wang et al., 2021) plots of the FFHQ and AFHQ experiments. Also, we have analyzed the found clusters using violin plots to visualize normalized distances between data points within each cluster. The plots also suggest that the first embedding can cluster the points more strongly compared to the second embedding. Also, we ran the K-means clustering algorithm 50 times on each of the embedding's features and computed the averaged (across the 50 runs) Adjusted Mutual Information (AMI) (Vinh et al., 2009) between the K-means labels and the SPEC-identified labels. The results indicate that the first embedding aligns more strongly with K-Means labels.

Furthermore, to highlight clustering differences between embeddings, we conducted a sanity check on two of the top five SPEC clusters from the DINOv2 - CLIP on AFHQ. We computed the center of the top four images in each cluster in both DINOv2 and CLIP embeddings. Then, we calculated the cosine similarity between the center and a set of eight test images: four additional images from the same cluster and four random images that do not belong to the cluster. As shown in Figure 16, DINOv2 well separates the cluster images from random samples, assigning the highest similarity scores to cluster-specific samples while keeping random samples significantly lower. However, in CLIP, some random images rank higher in similarity than

the cluster-specific samples. A similar experiment was performed on SPEC clusters from the DINOv2 - CLIP on the FFHQ dataset (Figure 17). Details are in Appendix B.5.

To further evaluate SPEC's performance in comparing embeddings, we apply a typographic attack on CLIP embeddings. As studied by Materzynska et al. (2022), CLIP prioritizes text added to a custom dataset over the image content. We selected 10 classes from the ImageNet-1K dataset and overlaid different text labels directly onto the images. The top four SPEC-identified clusters are presented in Figure 3, where we observe that CLIP clusters are based on the overlaid text, whereas DINOv2 clusters them based on visual features. We also compared CLIP and DINOv2 embeddings under the same settings on the ImageWoof dataset (Howard, 2019), which consists of various dog breeds from ImageNet-1K. SPEC principal clusters show that DINOv2 primarily clusters images based on dog breeds, whereas CLIP groups them based on the animals' gestures. Additional details are provided in Figure 15 of Appendix B.4.

**SPEC comparison of embeddings on different image and text datasets.** To check the performance of SPEC on text embeddings, we generated 10K samples from GPT-4o across different categories, including profession, object, gender, and emotion. We compared CLIP and RoBERTa text embeddings in Figure 4 and observed that the top four clusters in CLIP focused on objects in sentences, while RoBERTa clustered based on profession and gender. We

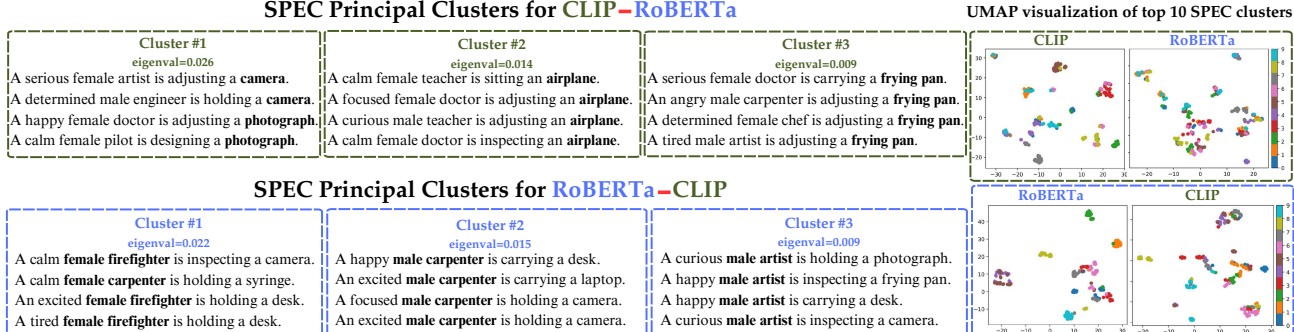

*Figure 4.* Top 4 SPEC-identified clusters by comparing CLIP and RoBERTa text embeddings on a dataset generated from GPT-4o.

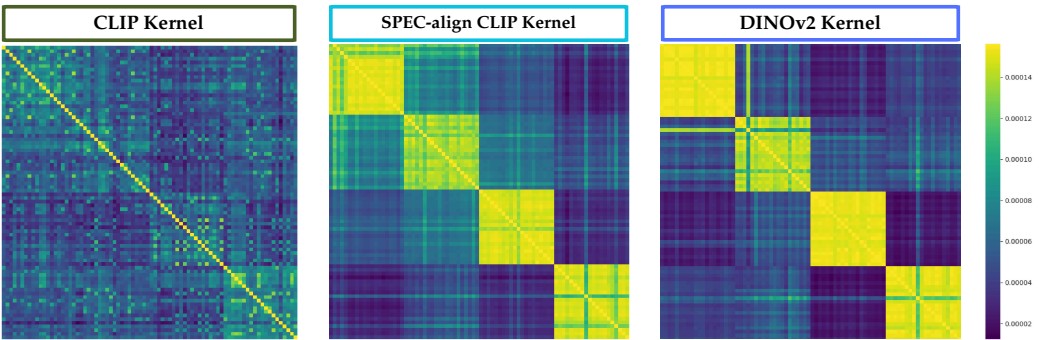

*Figure 5.* Comparison of Kernel matrices after using SPEC-align to match the sample clusters of CLIP to DINOv2.

also noted from the UMAP visualization of the samples that the top clusters of each embedding are not well clustered by the other, indicating that they focus on different aspects of the sentences. We also compared CLIP with E5 and observed the same results in Figure 13 of the Appendix B.3. In Figure 18, we compared different image embeddings on the MS-COCO 2017 training set with 120K samples which we discuss in the Appendix B.4.

**Aligning embeddings using SPEC-align** In this section, we discuss how to use the SPEC-align method to align the differential kernel covariance of two embeddings. As shown in the comparison of CLIP and DINOv2 in Figures 3 and 15, DINOv2 captures certain clusters that CLIP fails to distinguish. To improve CLIP's performance, we aligned it with DINOv2 using the ImageNet training set. Specifically, we added an alignment term to the CLIP loss, as formulated in (6), and computed the gradient using (7). Learning parameters are detailed in Appendix B.6, and additional results using Remark 2 are provided in Appendix B.1.

We provide the kernel matrices for the four clusters in this experiment in Figure 5, corresponding to the results in Figure 3. Notably, the SPEC-aligned CLIP kernel captures the top four clusters based on image content rather than the overlaid text labels. The clusters and their UMAP visualizations are shown in Figure 26. To further evaluate

SPEC-align's performance, we conducted an experiment similar to (Oquab et al., 2024), where feature quality was assessed by training a simple classifier on a frozen backbone without fine-tuning its weights. In this setting, SPEC-align CLIP achieved 73.93% top-1 accuracy on ImageNet-1K, outperforming the standard CLIP model, which reached 67.20%. For reference, DINOv2 achieved 78.99% on the same task, indicating that SPEC-align brings CLIP substantially closer to DINOv2 performance.

## 7. Conclusion

In this paper, we proposed the spectral SPEC approach to the comparison of embedding maps. The SPEC method aims to identify groups of samples clustered by one embedding model which is not grouped by another model. We formulated a scalable algorithm with $O(n)$ computations to apply SPEC to a dataset of size $n$. We also discussed the application of SPEC for measuring the mismatches of two embeddings and their alignment. We note that the SPEC approach operates based on the assumption that the differently clustered samples can be detected by the spectral method. Extending the clustering-based approach to non-spectral clustering frameworks will be interesting for future exploration. In addition, extending the framework to compare cross-modal embeddings such as CLIP and BLIP is a future direction for this work.

## Acknowledgments

The work of Farzan Farnia is partially supported by a grant from the Research Grants Council of the Hong Kong Special Administrative Region, China, Project 14209920, and is partially supported by CUHK Direct Research Grants with CUHK Project No. 4055164 and 4937054. Finally, the authors sincerely thank the anonymous reviewers for their useful feedback and constructive suggestions.

## Impact Statement

This paper presents work whose goal is to advance the field of Machine Learning. There are many potential societal consequences of our work, none which we feel must be specifically highlighted here.

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

# A. Proofs

## A.1. Proof of Theorem 1

For simplicity, we adopt the following notations in this proof:

$$K_{11}^{(1)} := \frac{1}{n}K_{\psi_1}[\mathcal{I}, \mathcal{I}], \quad K_{12}^{(1)} := \frac{1}{n}K_{\psi_1}[\mathcal{I}, \mathcal{I}^c], \quad K_{22}^{(1)} := \frac{1}{n}K_{\psi_1}[\mathcal{I}^c, \mathcal{I}^c]$$

$$K_{11}^{(2)} := \frac{1}{n}K_{\psi_2}[\mathcal{I}, \mathcal{I}], \quad K_{12}^{(2)} := \frac{1}{n}K_{\psi_2}[\mathcal{I}, \mathcal{I}^c], \quad K_{22}^{(2)} := \frac{1}{n}K_{\psi_2}[\mathcal{I}^c, \mathcal{I}^c]$$

$$\Lambda_{11} := \Lambda_{\psi_1,\psi_2}[\mathcal{I}, \mathcal{I}], \quad \Lambda_{12} := \Lambda_{\psi_1,\psi_2}[\mathcal{I}, \mathcal{I}^c], \quad \Lambda_{22} := \Lambda_{\psi_1,\psi_2}[\mathcal{I}^c, \mathcal{I}^c]$$

As a result, the following holds by definition:

$$\Lambda_{11} = K_{11}^{(1)} - K_{11}^{(2)}, \quad \Lambda_{12} = K_{12}^{(1)} - K_{12}^{(2)}, \quad \Lambda_{22} = K_{22}^{(1)} - K_{22}^{(2)}$$

According to Condition 1, we know the Frobenius norm bound $\|K_{12}^{(1)}\|_F \leq \epsilon_1$. Also, due to Condition 2, we know the $\ell_2$-operator norm bound $\|K_{11}^{(2)}\|_2 \leq \epsilon_2$. Note that the matrix $\frac{1}{n}K_{\psi_2}$ is positive semi-definite (PSD). Therefore, the Schur complement of its block representation following indices in $\mathcal{I}$ and $\mathcal{I}^c = \{1, \ldots, n\} - \mathcal{I}$ must be a PSD matrix, i.e.

$$K_{22}^{(2)} - K_{12}^{(2)\top}K_{11}^{(2)-1}K_{12}^{(2)} \succeq \mathbf{0}.$$

Therefore, the above Schur complement has a non-negative trace, implying that

$$\text{Tr}\left(K_{22}^{(2)} - K_{12}^{(2)\top}K_{11}^{(2)-1}K_{12}^{(2)}\right) \geq 0 \implies \text{Tr}\left(K_{12}^{(2)\top}K_{11}^{(2)-1}K_{12}^{(2)}\right) \leq \text{Tr}\left(K_{22}^{(2)}\right).$$

Therefore, we will have the following:

$$
\begin{aligned}
1 &= \text{Tr}\left(\frac{1}{n}K_{\psi_2}\right) \\
&\geq \text{Tr}\left(K_{22}^{(2)}\right) \\
&\geq \text{Tr}\left(K_{12}^{(2)\top}K_{11}^{(2)-1}K_{12}^{(2)}\right) \\
&\geq \text{Tr}\left(K_{12}^{(2)\top}\left(\frac{1}{\lambda_{\max}(K_{11}^{(2)})}I\right)K_{12}^{(2)}\right) \\
&\geq \frac{1}{\lambda_{\max}(K_{11}^{(2)})}\text{Tr}\left(K_{12}^{(2)\top}K_{12}^{(2)}\right) \\
&= \frac{1}{\|K_{11}^{(2)}\|_2}\text{Tr}\left(K_{12}^{(2)\top}K_{12}^{(2)}\right) \\
&= \frac{1}{\|K_{11}^{(2)}\|_2}\|K_{12}^{(2)}\|_F^2
\end{aligned}
$$

The above means that Condition 2 implies

$$\left\|K_{12}^{(2)}\right\|_F^2 \leq \left\|K_{11}^{(2)}\right\|_2 \leq \epsilon_2.$$

As a result, we can apply Young's inequality to show that

$$\|\Lambda_{12}\|_F^2 \leq 2\left(\left\|K_{12}^{(2)}\right\|_F^2 + \left\|K_{12}^{(1)}\right\|_F^2\right) \leq 2(\epsilon_1^2 + \epsilon_2).$$

Therefore, we will have the following:

$$\left\|\Lambda_{\psi_1,\psi_2} - \underbrace{\begin{bmatrix}\Lambda_{11} & \mathbf{0} \\ \mathbf{0} & \Lambda_{22}\end{bmatrix}}_{\widetilde{\Lambda}_{\psi_1,\psi_2}}\right\|_F^2 = 2\|\Lambda_{12}\|_F^2 \leq 4(\epsilon_1^2 + \epsilon_2).$$

Now, since $\Lambda_{\psi_1,\psi_2}$ is a symmetric matrix, we can apply spectral decomposition to write it as $\Lambda_{\psi_1,\psi_2} = V\mathrm{diag}(\boldsymbol{\lambda})V^\top$ where every row $\mathbf{v}_i$ of matrix $V$ is an eigenvector of $\Lambda_{\psi_1,\psi_2}$ with corresponding eigenvalue $\lambda_i$, sorted as $\lambda_1 \geq \cdots \geq \lambda_n$. Note that the eigenvalues are real and sum up to $0$ as the trace of $\Lambda_{\psi_1,\psi_2}$ is zero. Then, we can write:

$$
\begin{aligned}
\sum_{i=1}^n \big\|\widetilde{\Lambda}_{\psi_1,\psi_2}\mathbf{v}_i - \Lambda_{\psi_1,\psi_2}\mathbf{v}_i\big\|_2^2 &= \sum_{i=1}^n \big\|\big(\widetilde{\Lambda}_{\psi_1,\psi_2} - \Lambda_{\psi_1,\psi_2}\big)\mathbf{v}_i\big\|_2^2 \\
&= \sum_{i=1}^n \mathbf{v}_i^\top\big(\widetilde{\Lambda}_{\psi_1,\psi_2} - \Lambda_{\psi_1,\psi_2}\big)^\top\big(\widetilde{\Lambda}_{\psi_1,\psi_2} - \Lambda_{\psi_1,\psi_2}\big)\mathbf{v}_i \\
&= \sum_{i=1}^n \mathrm{Tr}\Big(\mathbf{v}_i^\top\big(\widetilde{\Lambda}_{\psi_1,\psi_2} - \Lambda_{\psi_1,\psi_2}\big)^\top\big(\widetilde{\Lambda}_{\psi_1,\psi_2} - \Lambda_{\psi_1,\psi_2}\big)\mathbf{v}_i\Big) \\
&= \sum_{i=1}^n \mathrm{Tr}\Big(\mathbf{v}_i\mathbf{v}_i^\top\big(\widetilde{\Lambda}_{\psi_1,\psi_2} - \Lambda_{\psi_1,\psi_2}\big)^\top\big(\widetilde{\Lambda}_{\psi_1,\psi_2} - \Lambda_{\psi_1,\psi_2}\big)\Big) \\
&= \mathrm{Tr}\Big(\big(\sum_{i=1}^n \mathbf{v}_i\mathbf{v}_i^\top\big)\big(\widetilde{\Lambda}_{\psi_1,\psi_2} - \Lambda_{\psi_1,\psi_2}\big)^\top\big(\widetilde{\Lambda}_{\psi_1,\psi_2} - \Lambda_{\psi_1,\psi_2}\big)\Big) \\
&= \mathrm{Tr}\Big(\big(\widetilde{\Lambda}_{\psi_1,\psi_2} - \Lambda_{\psi_1,\psi_2}\big)^\top\big(\widetilde{\Lambda}_{\psi_1,\psi_2} - \Lambda_{\psi_1,\psi_2}\big)\Big) \\
&= \big\|\widetilde{\Lambda}_{\psi_1,\psi_2} - \Lambda_{\psi_1,\psi_2}\big\|_F^2 \\
&\leq 4\big(\epsilon_1^2 + \epsilon_2\big)
\end{aligned}
$$

As a result, we can write

$$
\begin{aligned}
4\big(\epsilon_1^2 + \epsilon_2\big) &\geq \sum_{i=1}^n \big\|\widetilde{\Lambda}_{\psi_1,\psi_2}\mathbf{v}_i - \Lambda_{\psi_1,\psi_2}\mathbf{v}_i\big\|_2^2 \\
&= \sum_{i=1}^n \big\|\widetilde{\Lambda}_{\psi_1,\psi_2}\mathbf{v}_i - \lambda_i\mathbf{v}_i\big\|_2^2 \\
&= \sum_{i=1}^n \big\|\Lambda_{11}\mathbf{v}_i[\mathcal{I}] - \lambda_i\mathbf{v}_i[\mathcal{I}]\big\|_2^2 + \big\|\Lambda_{22}\mathbf{v}_i[\mathcal{I}^c] - \lambda_i\mathbf{v}_i[\mathcal{I}^c]\big\|_2^2 \\
&\geq \sum_{i=1}^n \big(\lambda_i - \lambda_i^{\mathcal{I}}\big)^2\big\|\mathbf{v}_i[\mathcal{I}]\big\|_2^2 + \big(\lambda_i - \lambda_i^{\mathcal{I}^c}\big)^2\big\|\mathbf{v}_i[\mathcal{I}^c]\big\|_2^2.
\end{aligned}
$$

In the above, the last inequality holds as we know for every PSD matrix $A$ and vector $\mathbf{v}$, we have $\|A\mathbf{v} - \lambda\mathbf{v}\|_2 \geq |\lambda_j - \lambda|\,\|\mathbf{v}\|_2$, where $\lambda_j$ is the eigenvalue of $A$ with the minimum absolute difference $|\lambda_j - \lambda|$. Therefore, the proof of Theorem 1 is complete.

### A.2. Proof of Proposition 1

Note that we can write $\Lambda_{\psi_1,\psi_2}$ using the following matrix multiplication:

$$
\Lambda_{\psi_1,\psi_2} = \frac{1}{n}\begin{bmatrix} \Phi_{\psi_1} & \Phi_{\psi_2} \end{bmatrix}\begin{bmatrix} \Phi_{\psi_1}^\top \\ -\Phi_{\psi_2}^\top \end{bmatrix}
$$

In the above, we define $\Phi_{\psi_1} \in \mathbb{R}^{n \times d_1}$ to be the embedding of dataset $x_1, \ldots, x_n$ with embedding map $\psi_1$, i.e., its $i$th row will be $\phi_1(\psi_1(x_i))$, and similarly we let $\Phi_{\psi_2} \in \mathbb{R}^{n \times d_2}$ to be the embedding of dataset with embedding map $\psi_2$ with its $i$th row being $\phi_2(\psi_2(x_i))$. Therefore, if we define $A = \begin{bmatrix} \Phi_{\psi_1} & \Phi_{\psi_2} \end{bmatrix}$ and $B = \frac{1}{n}\begin{bmatrix} \Phi_{\psi_1}^\top & -\Phi_{\psi_2}^\top \end{bmatrix}^\top$, then we will have $\Lambda_{\psi_1,\psi_2} = AB$.

On the other hand, we know that for every matrix $A \in \mathbb{R}^{n \times (d_1+d_2)}$ and $B \in \mathbb{R}^{(d_1+d_2) \times n}$, $AB$ and $BA$ share the same

non-zero eigenvalues. In this case, the matrix $BA$ sharing the non-zero eigenvalues with $\Lambda_{\psi_1,\psi_2} = AB$ can be calculated as

$$
\begin{aligned}
BA &= \frac{1}{n} \begin{bmatrix} \Phi_{\psi_1}^\top \\ -\Phi_{\psi_2}^\top \end{bmatrix} \begin{bmatrix} \Phi_{\psi_1} & \Phi_{\psi_2} \end{bmatrix} \\
&= \begin{bmatrix} \frac{1}{n}\Phi_{\psi_1}^\top \Phi_{\psi_1} & \frac{1}{n}\Phi_{\psi_1}^\top \Phi_{\psi_2} \\ -\frac{1}{n}\Phi_{\psi_2}^\top \Phi_{\psi_1} & -\frac{1}{n}\Phi_{\psi_2}^\top \Phi_{\psi_2} \end{bmatrix} \\
&= \begin{bmatrix} C_{\psi_1} & C_{\psi_1,\psi_2} \\ -C_{\psi_1,\psi_2}^\top & -C_{\psi_2} \end{bmatrix} \\
&= \Gamma_{\psi_1,\psi_2}.
\end{aligned}
$$

In addition, for every eigenvector $\mathbf{v}$ (corresponding to a non-zero eigenvalue) of $\Gamma_{\psi_1,\psi_2} = BA$, we have that $\mathbf{u} = A\mathbf{v}$ is an eigenvector of $\Lambda_{\psi_1,\psi_2} = AB$ which is

$$
\mathbf{u} = \begin{bmatrix} \Phi_{\psi_1} & \Phi_{\psi_2} \end{bmatrix} \mathbf{v} = \begin{bmatrix} \phi_1(\psi_1(x_1)) & \phi_2(\psi_2(x_1)) \\ \vdots & \vdots \\ \phi_1(\psi_1(x_n)) & \phi_2(\psi_2(x_n)) \end{bmatrix} \mathbf{v}
$$

Therefore, the proof is complete.

### A.3. Computation of Eigenvectors and Eigenvalues of $\Gamma_{\psi_1,\psi_2}$

Considering the notations in the previous proof, note that the defined differential covariance matrix $\Gamma_{\psi_1,\psi_2}$ can be written as follows:

$$
\Gamma_{\psi_1,\psi_2} = \begin{bmatrix} C_{\psi_1} & C_{\psi_1,\psi_2} \\ -C_{\psi_1,\psi_2}^\top & -C_{\psi_2} \end{bmatrix} = \underbrace{\begin{bmatrix} I_{d_1 \times d_1} & \mathbf{0}_{d_1 \times d_2} \\ \mathbf{0}_{d_2 \times d_1} & -I_{d_2 \times d_2} \end{bmatrix}}_{\text{Matrix } D} \underbrace{\begin{bmatrix} C_{\psi_1} & C_{\psi_1,\psi_2} \\ C_{\psi_1,\psi_2}^\top & C_{\psi_2} \end{bmatrix}}_{\text{Matrix } C}
$$

In the above, $D$ is a diagonal matrix with $\pm 1$ diagonal entries, and $C$ represents the joint kernel covariance matrix of the concatenation of the embedding vectors $[\phi_1(\psi_1(x)), \phi_2(\psi_2(x))]$ that is a PSD matrix. Since $C$ is a $(d_1 + d_2) \times (d_1 + d_2)$ symmetric PSD matrix, we can apply the Cholesky decomposition to write $C = Z^\top Z$ for an upper-triangular matrix $Z \in \mathbb{R}^{(d_1+d_2) \times (d_1+d_2)}$. Therefore, we have $\Gamma_{\psi_1,\psi_2} = DZ^\top Z$. While $\Gamma_{\psi_1,\psi_2} = DZ^\top Z$ is not a symmetric matrix, we note that the multiplication-order-flipped $\Theta = ZDZ^\top \in \mathbb{R}^{(d_1+d_2) \times (d_1+d_2)}$ will be a symmetric matrix. Therefore, the symmetric matrix $\Theta$ shares the same non-zero eigenvalues with $\Gamma_{\psi_1,\psi_2}$, and for each eigenvector $w$ of $\Theta$, the vector $u = DZ^\top w$ will be the corresponding eigevector of $\Gamma_{\psi_1,\psi_2}$. To conclude, one can perform the eigendecomposition for the symmetric matrix $\Theta$ and then convert the non-zero eigenvalues and eigenvectors to compute those of $\Gamma_{\psi_1,\psi_2}$, resulting in the following procedure:

1. Form the PSD matrix $C = \begin{bmatrix} C_{\psi_1} & C_{\psi_1,\psi_2} \\ C_{\psi_1,\psi_2}^\top & C_{\psi_2} \end{bmatrix}$

2. Apply Cholesky decomposition to write $C = Z^\top Z$ for an upper-triangular matrix $Z \in \mathbb{R}^{(d_1+d_2) \times (d_1+d_2)}$

3. Compute the symmetric matrix $\Theta = Z \begin{bmatrix} I_{d_1 \times d_1} & \mathbf{0}_{d_1 \times d_2} \\ \mathbf{0}_{d_2 \times d_1} & -I_{d_2 \times d_2} \end{bmatrix} Z^\top$

4. Perform eigendecomposition on symmetric matrix $\Theta$ to compute $d_1 + d_2$ pairs of eigenvalues and eigenvectors $(\lambda_i, w_i)_{i=1}^{d_1+d_2}$

5. For each $i \in \{1, \ldots d_1 + d_2\}$, compute $u_i = \begin{bmatrix} I_{d_1 \times d_1} & \mathbf{0}_{d_1 \times d_2} \\ \mathbf{0}_{d_2 \times d_1} & -I_{d_2 \times d_2} \end{bmatrix} Z^\top w_i$

### A.4. Proof of Theorem 2

As stated in the theorem, we consider independent random Fourier features $\omega_1, \ldots, \omega_m \sim \widehat{\kappa}$ where the proxy feature map is:

$$\widehat{\phi}(x) = \frac{1}{\sqrt{m}}\Big[\cos(\omega_1^\top x), \sin(\omega_1^\top x), ., ., \cos(\omega_m^\top x), \sin(\omega_m^\top x)\Big]$$

Based on the assumption, the shift-invariant kernel $k(x, y) = \kappa(x - y)$ is normalized where $k(x, x) = 1$ for every $x \in \mathcal{X}$. Therefore, using the Fourier synthesis equation $k(x, y) = \mathbb{E}_{\omega \sim \widehat{\kappa}}\big[\cos(\omega^\top(x - y))\big] = \mathbb{E}_{\omega \sim \widehat{\kappa}}\big[\cos(\omega^\top x)\cos(\omega^\top y) + \sin(\omega^\top x)\sin(\omega^\top y)\big]$. The RFF-proxy kernel function can be viewed as

$$\widehat{k}(x, y) = \frac{1}{m}\sum_{i=1}^{m}\cos(\omega_i^\top(x - y)).$$

As a result, if we consider kernel matrix $K_{\psi_1, \omega_i}$ where $k_{\psi_1, \omega_i}(x, y) = \cos(\omega_i^\top(\psi_1(x) - \psi_1(y)))$, we can simplify the proxy kernel matrix as

$$\frac{1}{n}\widehat{K}_{\psi_1} = \frac{1}{m}\sum_{i=1}^{m}\frac{1}{n}K_{\psi_1, \omega_i}$$

where we note that $\mathbb{E}_{\omega_i \sim p_\omega}\big[\frac{1}{n}K_{\psi_1, \omega_i}\big] = \frac{1}{n}K_{\psi_1}$ as $\omega$ is drawn from the Fourier transform $\widehat{\kappa}$.

Also, $\|\frac{1}{n}K_{\psi_1, \omega_i}\|_F \leq 1$ holds, because the kernel function is assumed to be normalized and $|k_{\psi_1}(x, y)| \leq 1$ for every $x, y$. Noting that the Frobenius norm $\|\cdot\|_F$ can be written as the Euclidean norm of the vectorized matrix, the application of Vector Bernstein inequality (Gross, 2011; Kohler & Lucchi, 2017) proves for any $0 \leq \epsilon \leq 2$:

$$\mathbb{P}\Big(\Big\|\frac{1}{m}\sum_{i=1}^{m}[\frac{1}{n}K_{\psi_1, \omega_i}] - \frac{1}{n}K_{\psi_1}\Big\|_F \geq \epsilon\Big) \leq \exp\Big(\frac{8 - m\epsilon^2}{32}\Big),$$

Therefore, we will have

$$\mathbb{P}\Big(\Big\|\frac{1}{n}\widehat{K}_{\psi_1} - \frac{1}{n}K_{\psi_1}\Big\|_F \geq \epsilon\Big) \leq \exp\Big(\frac{8 - m\epsilon^2}{32}\Big)$$

Similarly, we can show that for the embedding $\psi_2$ we will have

$$\mathbb{P}\Big(\Big\|\frac{1}{n}\widehat{K}_{\psi_2} - \frac{1}{n}K_{\psi_2}\Big\|_F \geq \epsilon\Big) \leq \exp\Big(\frac{8 - m\epsilon^2}{32}\Big)$$

Next, we note that

$$
\begin{aligned}
\sum_{i=1}^{n}\left\|\Lambda_{\psi_1,\psi_2}\widehat{\mathbf{v}}_i - \lambda_i\widehat{\mathbf{v}}_i\right\|_2^2 &= \sum_{i=1}^{n}\left\|\Lambda_{\psi_1,\psi_2}\widehat{\mathbf{v}}_i - \widehat{\Lambda}_{\psi_1,\psi_2}\widehat{\mathbf{v}}_i\right\|_2^2 \\
&= \sum_{i=1}^{n}\left\|\left(\Lambda_{\psi_1,\psi_2} - \widehat{\Lambda}_{\psi_1,\psi_2}\right)\widehat{\mathbf{v}}_i\right\|_2^2 \\
&= \sum_{i=1}^{n}\widehat{\mathbf{v}}_i^{\top}\left(\Lambda_{\psi_1,\psi_2} - \widehat{\Lambda}_{\psi_1,\psi_2}\right)^{\top}\left(\Lambda_{\psi_1,\psi_2} - \widehat{\Lambda}_{\psi_1,\psi_2}\right)\widehat{\mathbf{v}}_i \\
&= \sum_{i=1}^{n}\mathrm{Tr}\left(\widehat{\mathbf{v}}_i^{\top}\left(\Lambda_{\psi_1,\psi_2} - \widehat{\Lambda}_{\psi_1,\psi_2}\right)^{\top}\left(\Lambda_{\psi_1,\psi_2} - \widehat{\Lambda}_{\psi_1,\psi_2}\right)\widehat{\mathbf{v}}_i\right) \\
&= \sum_{i=1}^{n}\mathrm{Tr}\left(\left(\Lambda_{\psi_1,\psi_2} - \widehat{\Lambda}_{\psi_1,\psi_2}\right)^{\top}\left(\Lambda_{\psi_1,\psi_2} - \widehat{\Lambda}_{\psi_1,\psi_2}\right)\widehat{\mathbf{v}}_i\widehat{\mathbf{v}}_i^{\top}\right) \\
&= \mathrm{Tr}\left(\left(\Lambda_{\psi_1,\psi_2} - \widehat{\Lambda}_{\psi_1,\psi_2}\right)^{\top}\left(\Lambda_{\psi_1,\psi_2} - \widehat{\Lambda}_{\psi_1,\psi_2}\right)\left(\sum_{i=1}^{n}\widehat{\mathbf{v}}_i\widehat{\mathbf{v}}_i^{\top}\right)\right) \\
&= \mathrm{Tr}\left(\left(\Lambda_{\psi_1,\psi_2} - \widehat{\Lambda}_{\psi_1,\psi_2}\right)^{\top}\left(\Lambda_{\psi_1,\psi_2} - \widehat{\Lambda}_{\psi_1,\psi_2}\right)\right) \\
&= \left\|\Lambda_{\psi_1,\psi_2} - \widehat{\Lambda}_{\psi_1,\psi_2}\right\|_F^2 \\
&\le 2\left\|K_{\psi_1} - \widehat{K}_{\psi_1}\right\|_F^2 + 2\left\|K_{\psi_2} - \widehat{K}_{\psi_2}\right\|_F^2
\end{aligned}
$$

The last line in the above inequalities follow from Young's inequality showing that $\|A + B\|_F^2 \le 2\|A\|_F^2 + 2\|B\|_F^2$. Then, setting $\delta = 2\exp\left(\frac{8-m\epsilon^2}{32}\right)$ implying $\epsilon = \sqrt{\frac{32\log(2e^{1/4}/\delta)}{m}}$, we will have

$$
\mathbb{P}\left(\left\|\frac{1}{n}\widehat{K}_{\psi_1} - \frac{1}{n}K_{\psi_1}\right\|_F \le \sqrt{\frac{32\log(2e^{1/4}/\delta)}{m}}\right) \ge 1 - \frac{\delta}{2}, \quad \mathbb{P}\left(\left\|\frac{1}{n}\widehat{K}_{\psi_2} - \frac{1}{n}K_{\psi_2}\right\|_F \le \sqrt{\frac{32\log(2e^{1/4}/\delta)}{m}}\right) \ge 1 - \frac{\delta}{2}
$$

where by applying the union bound we can show

$$
\mathbb{P}\left(\left\|\frac{1}{n}\widehat{K}_{\psi_1} - \frac{1}{n}K_{\psi_1}\right\|_F \le \sqrt{\frac{32\log(2e^{1/4}/\delta)}{m}} \text{ and } \left\|\frac{1}{n}\widehat{K}_{\psi_2} - \frac{1}{n}K_{\psi_2}\right\|_F \le \sqrt{\frac{32\log(2e^{1/4}/\delta)}{m}}\right) \ge 1 - \delta
$$

which shows that

$$
\mathbb{P}\left(\sum_{i=1}^{n}\left\|\Lambda_{\psi_1,\psi_2}\widehat{\mathbf{v}}_i - \lambda_i\widehat{\mathbf{v}}_i\right\|_2^2 \le \frac{128\log(2e^{1/4}/\delta)}{m}\right) \ge 1 - \delta
$$

The above completes the theorem's proof given that $2e^{1/4} \approx 2.57 < 3$.

## A.5. Proof of Proposition 2

To show this statement, we leverage the fact that the eigenvalues of matrix $\Gamma_{\psi_{1,\theta},\psi_2}$ are real, as they are shared with the symmetric matrix $\Lambda_{\psi_1,\psi_2}$. Then, we can leverage the Jordan canonical form to write the matrix $\Gamma_{\psi_{1,\theta},\psi_2}$ as follows:

$$
\Gamma_{\psi_{1,\theta},\psi_2} = UJU^{-1}
$$

In the above, matrix $U \in \mathbb{R}^{d_1+d_2}$ includes the right generalized eigenvectors of matrix $\Gamma_{\psi_{1,\theta},\psi_2}$ as its rows, and $U^{-1}$ includes the left generalized eigenvectors of matrix $\Gamma_{\psi_{1,\theta},\psi_2}$ in its rows. Also, $J$ is the Jordan normal form containing one block matrix on the diagonal for every eigenvalue. Assuming that the eigenvalue with the maximum absolute value (which determines the spectral radius) has multiplicity 1, the Jordan canonical form has only one diagonal entry $\lambda_{\max}$ for the top eigenvalue, and so we can write the decomposition as:

$$
\Gamma_{\psi_{1,\theta},\psi_2} = U^{(-i_{\max})}J^{(-i_{\max})}U^{-1}{}^{(-i_{\max})} + \lambda_{\max}\mathbf{u}_{\text{left}}\mathbf{u}_{\text{right}}^{\top}
$$

| | Time (sec) | | | | | |
|---|---|---|---|---|---|---|
| Computation Method | $d = 1000$ | $d = 2000$ | $d = 4000$ | $d = 6000$ | $d = 8000$ | $d = 10000$ |
| Direct (Algorithm 1) | 13 | 35 | 72 | - | - | - |
| Cholesky Decomposition-based (Appendix A.3) | 11 | 22 | 33 | 47 | 64 | 85 |

*Table 1.* Time taken for Direct method and Cholesky Decomposition method at different values of $d$.

Due to the bi-orthogonality of $\mathbf{u}_{\text{left}}$, $\mathbf{u}_{\text{right}}$ with the rest of right and left generalized eigenvectors, respectively, we will have

$$\lambda_{\max}(\Gamma_{\psi_{1,\theta},\psi_2}) = \mathbf{u}_{\text{left}}^\top \Gamma_{\psi_{1,\theta},\psi_2} \mathbf{u}_{\text{right}}$$

Taking the partial derivative with respect to $\theta$ from the above identity proves the proposition.

## B. Additional Numerical Results

In this section, we provide additional numerical results on embedding comparisons and alignment using the SPEC framework, further illustrating its effectiveness in identifying clustering differences across various datasets and embedding methods.

### B.1. Cholesky decomposition-based method for eigenvectors and eigenvalues computation

As explained in Remark 1, the computation of the eigenvalues and eigenvectors of the differential covariance matrix $\Gamma_{\psi_1,\psi_2}$ can be efficiently reduced to the eigendecomposition of a symmetric matrix through the use of Cholesky decomposition. We followed the explanation in the Appendix A.3 and compared the runtime between Algorithm 1 and the Cholesky decomposition-based method on the Gaussian Kernel with different RFF. As shown in Table 1, the Cholesky method is more scalable and was able to handle $d = 10K$, whereas the vanilla implementation of Algorithm 1 using Pytorch's eig command (for general square matrices) could not efficiently scale beyond dimension $d = 5000$.

Furthermore, in Figures 6, 7, we present the top four SPEC-identified clusters comparing CLIP with DINOv2 using different values of Random Fourier Features (RFF) $r$, ranging from 1000 to 6000, with Cholesky decomposition applied to the MS-COCO dataset. We observe that for $r \leq 3000$, the clusters are not well-separated and show some inconsistencies. As $r$ increases, these inconsistencies diminish, and by $r = 6000$, the clusters are clean and well-separated. We also provide the t-SNE and UMAP representations of the top 10 SPEC-identified clusters for CLIP and DINOv2 for $r = 2000, 6000$. in Figures 8, 9.

### B.2. Ablation Study on the Comparison of Visualization Algorithms

To further validate that the SPEC method can distinguish between two embeddings, we supplemented the UMAP visualization in Figure 2 with additional techniques, PacMAP (Wang et al., 2021) and t-SNE. These visualizations assess how well the clusters identified by SPEC align across different dimensionality reduction methods. As shown in Figure 10, the alternative techniques confirm that SPEC correctly isolates distinct clusters in one embedding, whereas the other embedding fails to produce clean separations.

### B.3. Comparison of text embeddings

**Synthetic text dataset.** To evaluate the performance of SPEC on text embeddings, we generated a dataset of 10K text samples using GPT-4o, covering diverse categories such as profession, objects, gender, and emotions. We then applied SPEC to compare CLIP, RoBERTa, and E5 text embeddings. As shown in Figures 12 and 13, CLIP primarily clusters sentences based on objects mentioned in the text, whereas RoBERTa organizes them according to profession and gender. For instance, CLIP's first cluster consists of sentences related to cameras and photography, while RoBERTa's first cluster groups sentences about female firefighters and female carpenters. This suggests that CLIP embeddings do not cluster professions based on gender but excel at grouping objects, which aligns with its training focus. Additionally, the t-SNE visualization reveals that the principal clusters identified in one embedding are not well-separated in the other, indicating that each model captures different semantic aspects of the text. In addition to previous experiments, we used MS-COCO 2017 train set captions ( 120K samples) to compare RoBERTa and E5-L-V2 embeddings. As shown in Figure 14, we can observe that E5 managed to cluster captions where two or more animals are interacting.

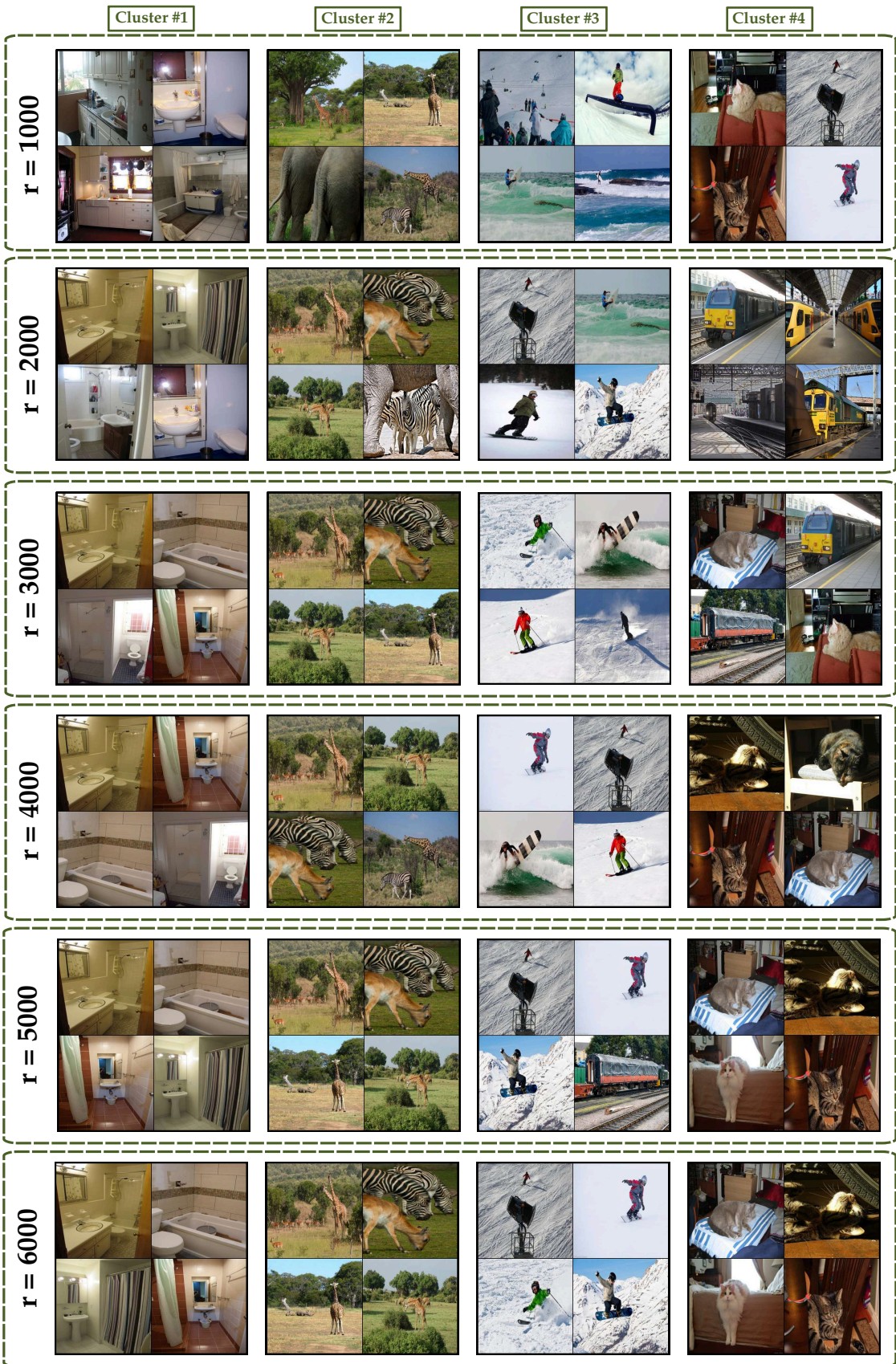

*Figure 6.* Top 4 SPEC-identified clusters comparing CLIP - DINOv2 with different Random Fourier features (RFF) r ranging from 1000 to 6000 using Cholesky decomposition on MS-COCO dataset.

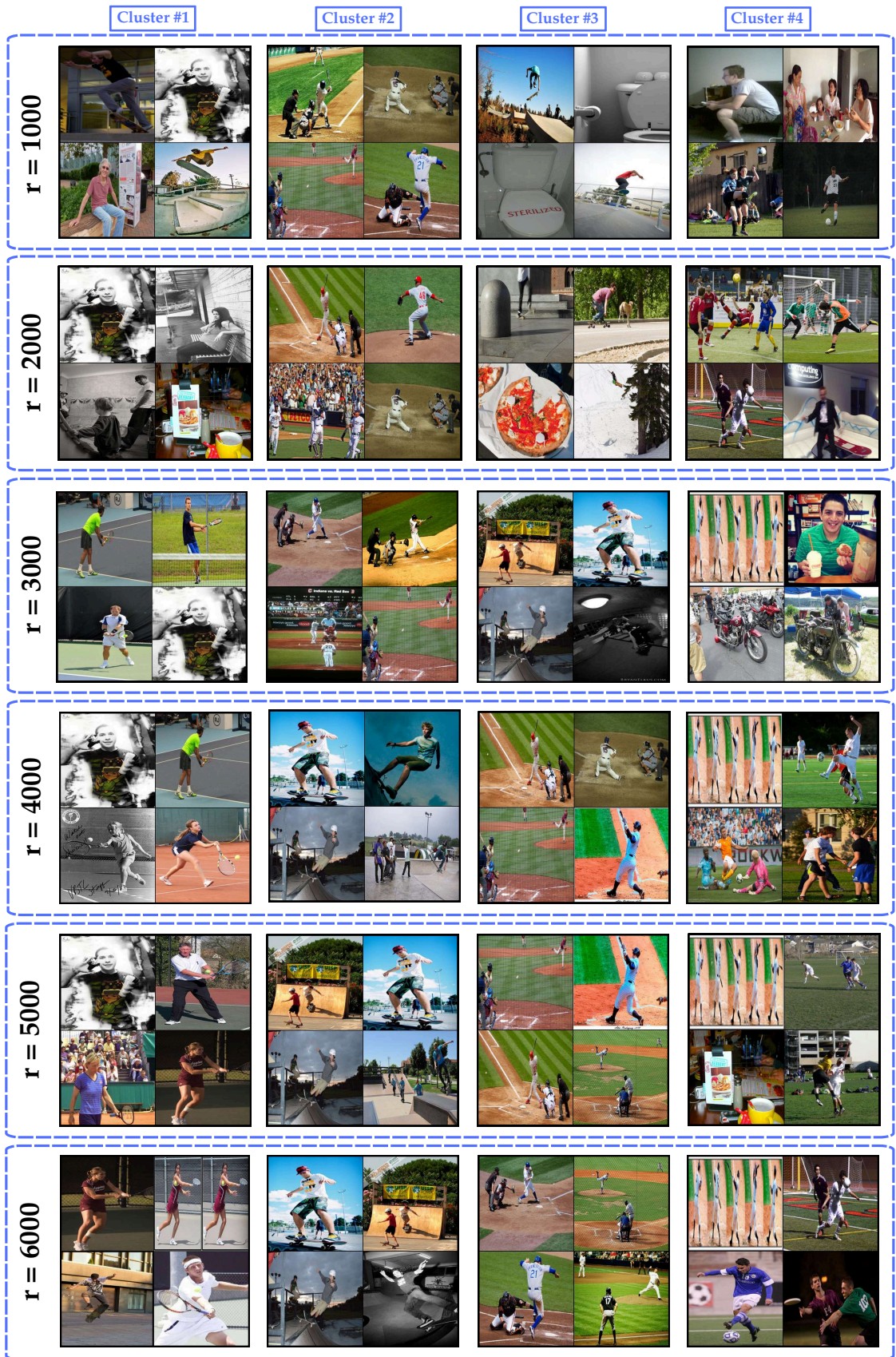

*Figure 7.* Top 4 SPEC-identified clusters comparing DINOv2 - CLIP with different Random Fourier features (RFF) r ranging from 1000 to 6000 using Cholesky decomposition on MS-COCO dataset.

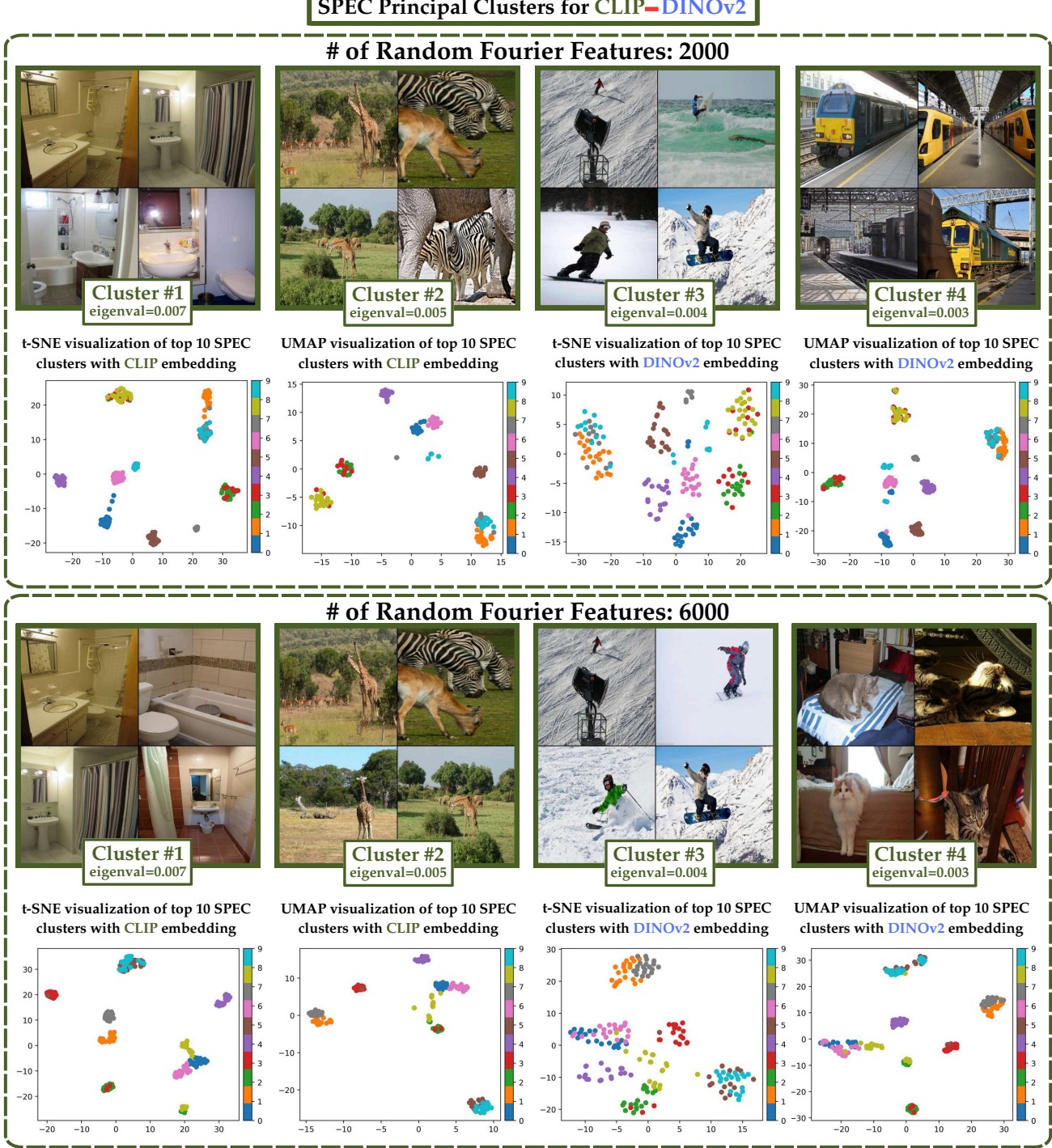

*Figure 8.* Top 4 SPEC-identified clusters comparing CLIP - DINOv2 embeddings on MS-COCO dataset using Cholesky decomposition with different Random Fourier Features (2000 and 6000). The second row shows the t-SNE and UMAP representation of the top 10 SPEC-identified clusters for each embedding.

Figure 9. Top 4 SPEC-identified clusters comparing DINOv2 - CLIP embeddings on MS-COCO dataset using Cholesky decomposition with different Random Fourier Features (2000 and 6000). The second row shows the t-SNE and UMAP representation of the top 10 SPEC-identified clusters for each embedding.

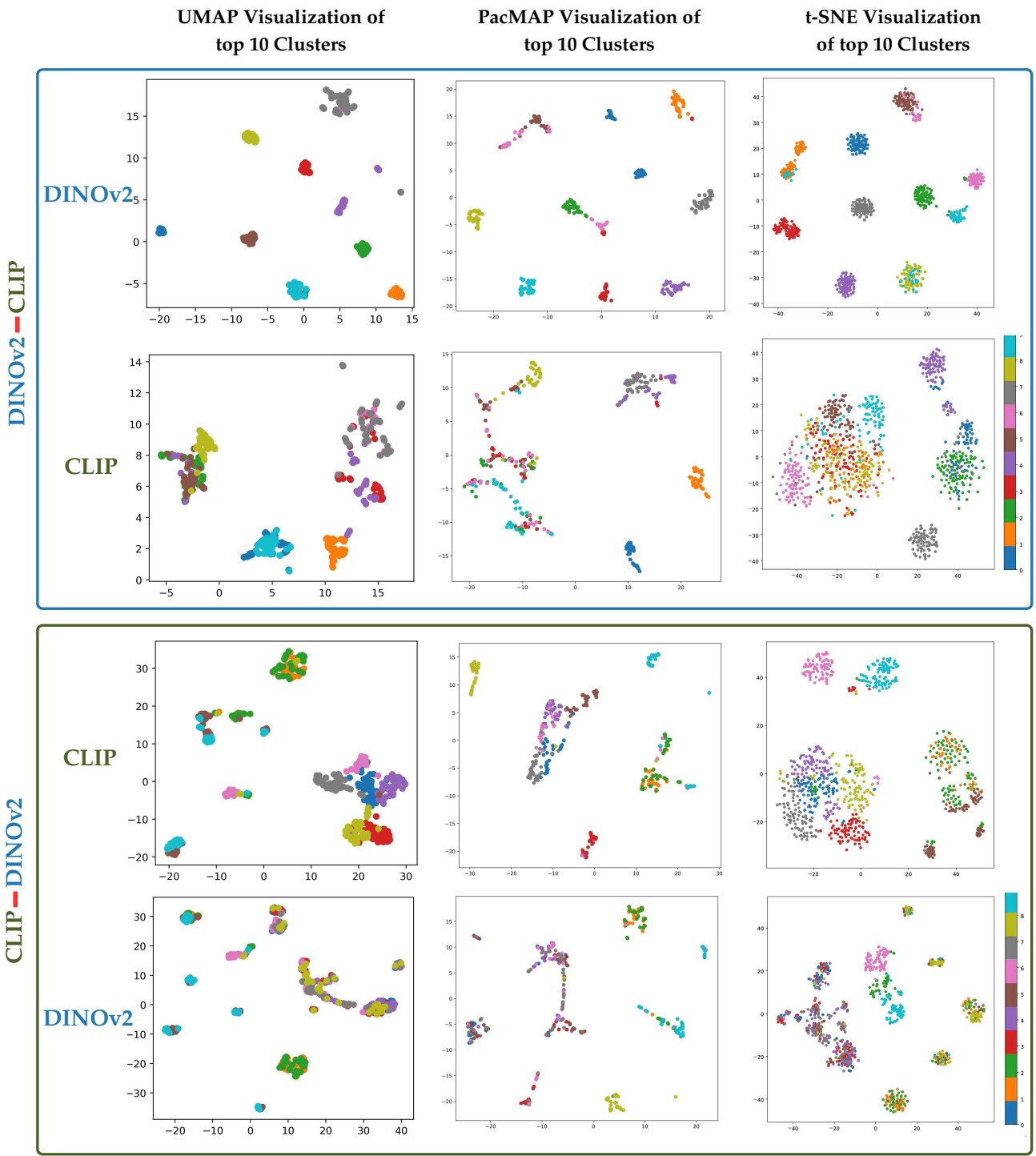

*Figure 10.* Comparison of SPEC-identified clusters across different visualization methods (PacMAP, t-SNE, UMAP) of Figure 2 experiment. One embedding shows clear cluster separation, while the other fails to distinguish groups.

The prompt used to generate the text dataset: *"You are an expert prompt optimizer for text-to-image models. Text-to-image models take a text prompt as input and generate images. Your task is to generate a prompt describing a person in [Profession], [Emotion], and [Gender] performing [Action] with [Object]. You can randomly choose the categories from the attributes: Professions: Chef, doctor, journalist, scientist, carpenter, engineer, pilot, artist, teacher, firefighter. Emotions: Excited, calm, angry, serious, curious, confident, focused, determined, happy, tired. Genders: Male, female. Actions: Designing, adjusting, sitting, crouching, climbing, carrying, holding, standing, inspecting, juggling. Objects: Camera, frying pan, painting, laptop, photograph, syringe, golden throne, desk, guitar, airplane."*

**Real world text dataset.** To further compare the text embeddings, we validated our approach on a large-scale real text dataset: WikiText-2 (Merity et al., 2016). We split the dataset into 10K samples, each containing 100 tokens. Then, we used SPEC to compare CLIP and RoBERTa embeddings. As shown in Figure 11. We observed that RoBERTa better clustered Military Operations & Infrastructure, Ecology & Species Biology, Historical Figures, and Music, while CLIP embeddings more strongly clustered Entertainment & Sports and Science.

In addition to t-SNE plots, we also examined the distribution of pairwise distances within each cluster to verify that one embedding successfully captured these clusters while the other was less inclined to do so. Also, we ran the K-means clustering algorithm 50 times on each of the embedding's features and computed the averaged (across the 50 runs) Normalized Mutual Information (NMI) between the K-means labels and the SPEC-identified labels. The results demonstrate that one embedding achieved considerably stronger alignment with KMeans labels.

## B.4. Comparison of image embeddings

To explore the capability of CLIP's image embedding, we analyze the top 4 SPEC-identified clusters comparing CLIP and DINOv2 embeddings on ImageNet-1k dog breeds in Figure 15, highlighting their different clustering strategies. CLIP primarily groups dogs based on their posture and gestures rather than their breed. For example, in cluster two, all dogs are standing, but they belong to different breeds. This suggests that CLIP focuses more on high-level visual features like body position and orientation. In contrast, DINOv2 forms clusters based on dog breeds, grouping visually similar dogs together regardless of their posture. The last row presents the t-SNE representation of the top 10 SPEC-identified clusters for each embedding, further illustrating their distinct clustering behaviors.

We analyzed the clustering behavior of different embeddings on 120K samples from the MS-COCO 2017 dataset and observed similar trends in how different models organize visual concepts in Figure 18. For instance, SWAV demonstrates a strong ability to cluster grid-like images, suggesting its emphasis on structural patterns in images. Meanwhile, CLIP excels at differentiating activities like surfing, capturing fine-grained semantic details that may not be as distinct in SWAV or DINOv2. However, CLIP struggles to cluster certain sports, such as tennis, as effectively as SWAV or DINOv2, highlighting its relative limitations in capturing specific action-based similarities. These findings further illustrate the varying strengths of different embeddings in organizing visual content.

We also analyzed SPEC on 70,000 samples from the FFHQ dataset. As observed in Figure 19, DINOv2 better distinguishes images where two people are present, with one appearing incompletely, as a cluster. In contrast, CLIP is more effective at identifying children as a distinct group.

## B.5. Comparing similarity ranking for SPEC clusters

In Figure 16, the leftmost images show the top 4 samples of SPEC-identified clusters on AFHQ. The first and second clusters correspond to black cats and black & white cats, respectively. For each cluster, cosine similarity is computed between the mean of these samples and both 4 additional cluster members (green-bordered) and 4 random images (red-bordered). Images are sorted left to right by similarity scores. The first row represents DINOv2, the second CLIP. Unlike DINOv2, CLIP ranks some random cats more similar to the cluster mean than the actual cluster members.

For the FFHQ dataset, as shown in Figure 17, the first cluster corresponds to people wearing graduation caps, and the second to people wearing sunglasses. While DINOv2 maintains a significant similarity gap, clearly distinguishing the clusters, CLIP assigns higher similarity to some samples that lack these defining features.

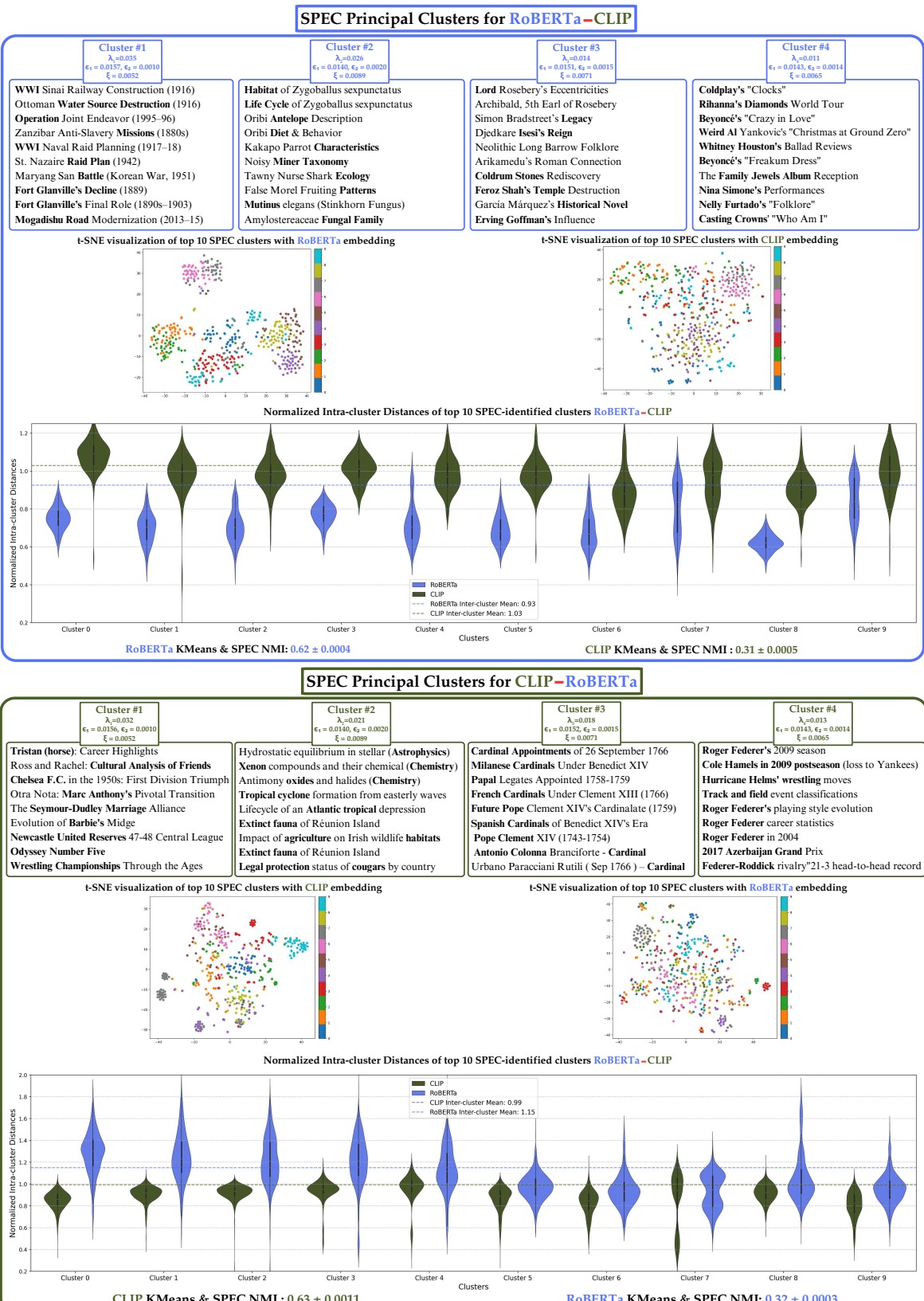

Figure 11. Top 4 SPEC-identified clusters by comparing CLIP and RoBERTa text embeddings on the WikiText-2 dataset with the visualization of the top 10 SPEC-identified clusters using t-SNE.

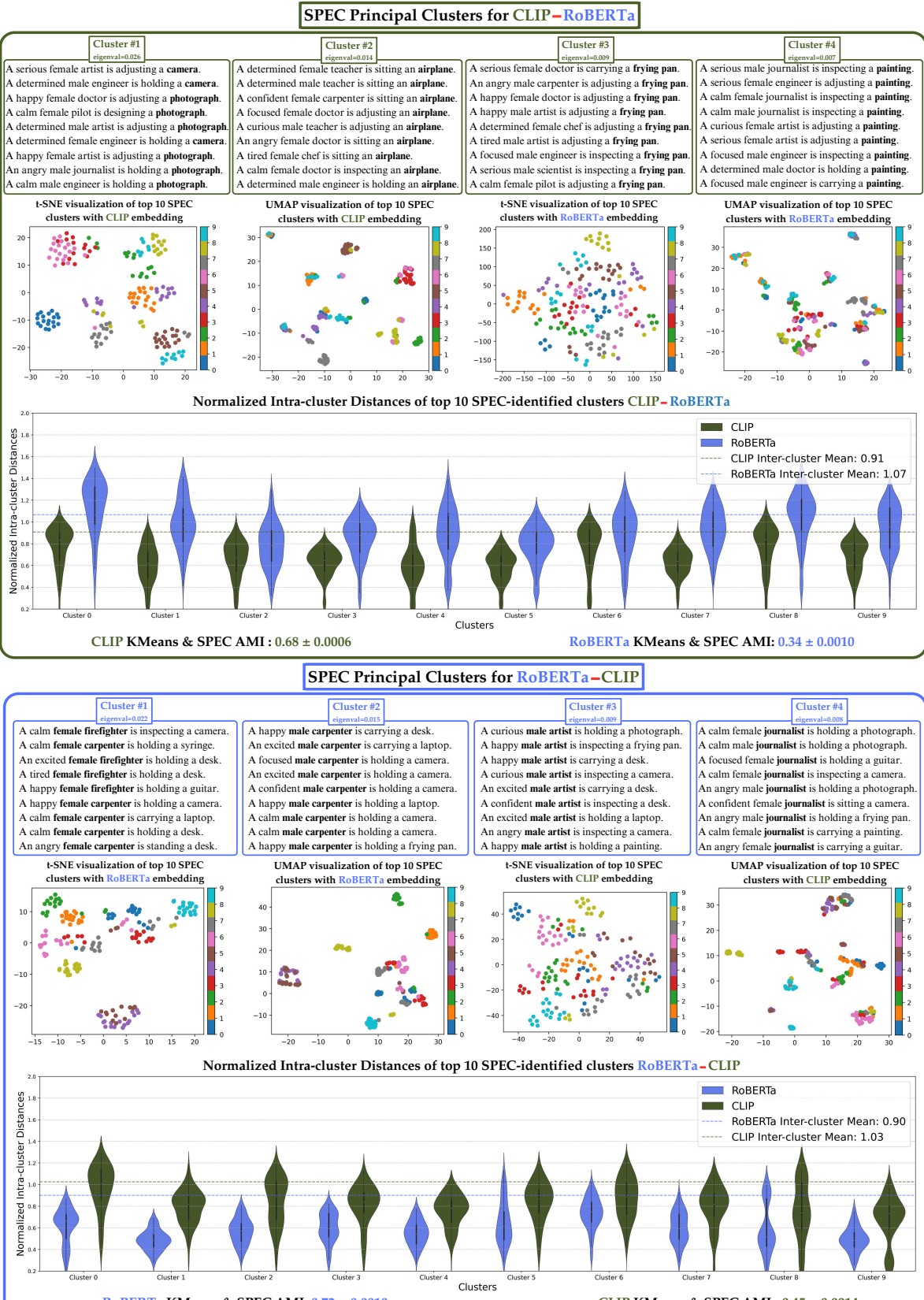

Figure 12. Top 4 SPEC-identified clusters by comparing CLIP and RoBERTa text embeddings on a dataset of 10K samples generated from GPT-4o with the visualization of the top 10 SPEC-identified clusters using t-SNE and UMAP.

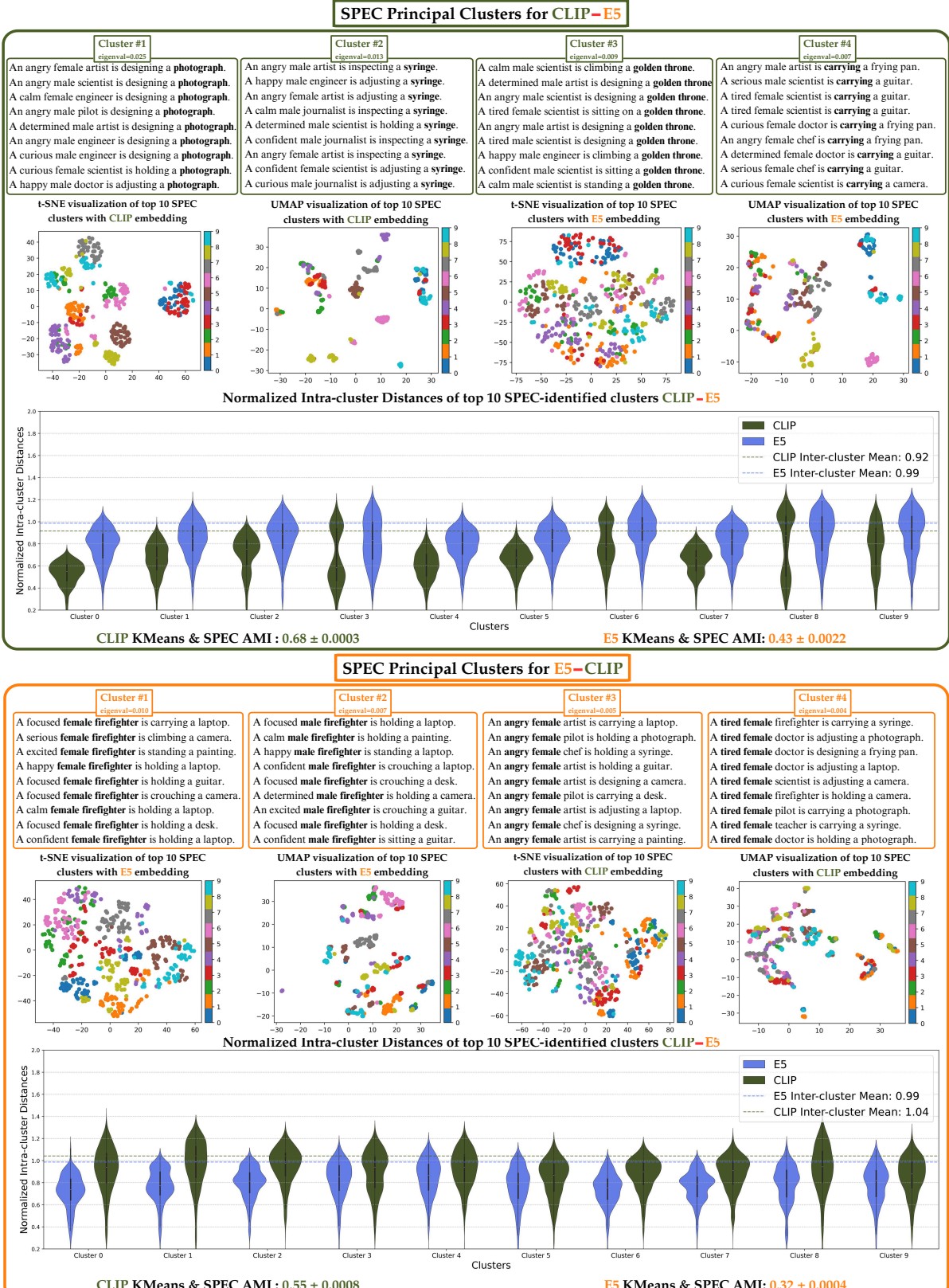

*Figure 13.* Top 4 SPEC-identified clusters by comparing CLIP and E5-Large-V2 text embeddings on a dataset of 10K samples generated from GPT-4o with the visualization of the top 10 SPEC-identified clusters using t-SNE.

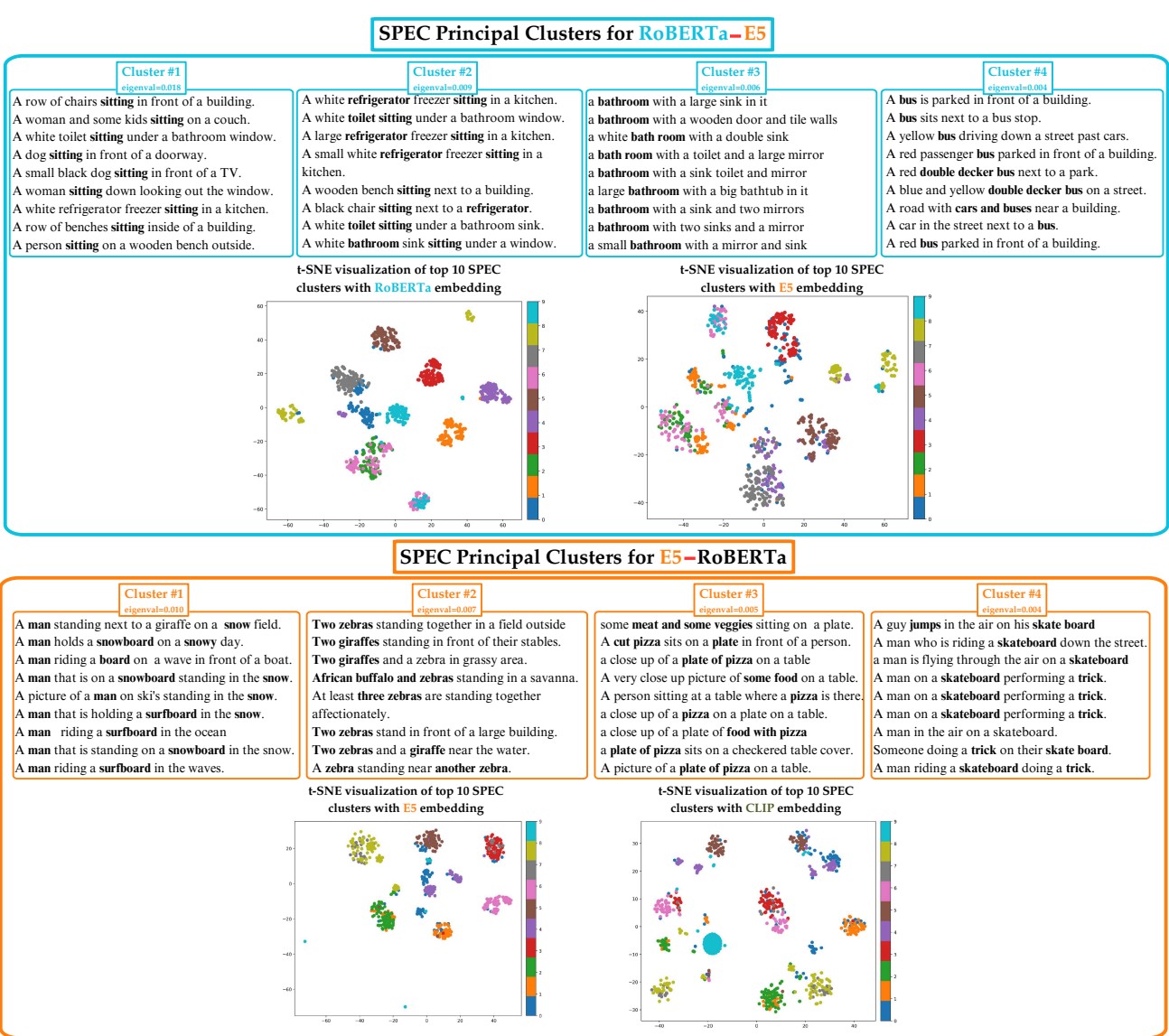

Figure 14. Top 4 SPEC-identified clusters by comparing RoBERTa and E5-Large-V2 text embeddings on MS-COCO 2017 train captions ( 120K prompts) with the visualization of the top 10 SPEC-identified clusters using t-SNE.

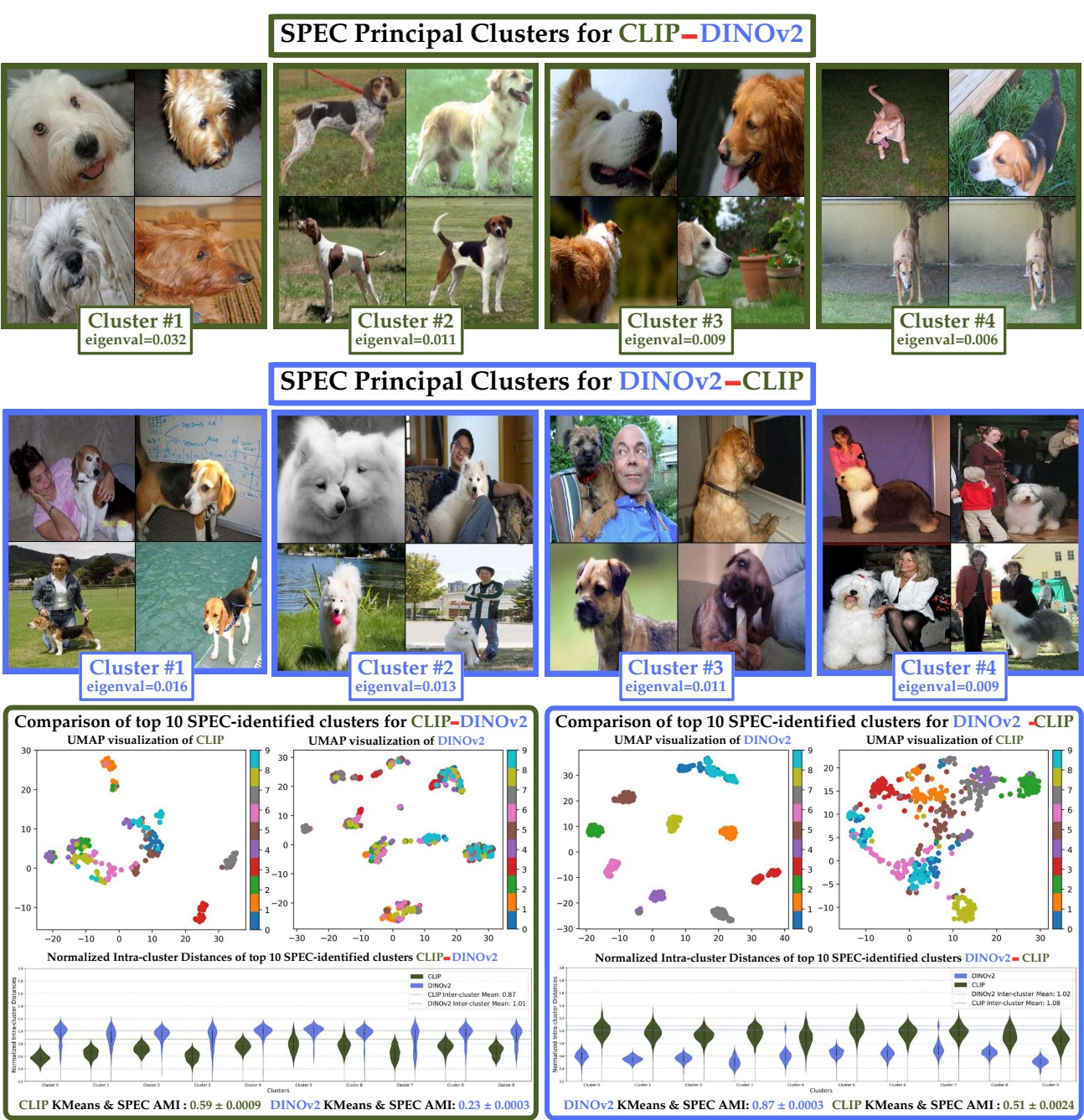

*Figure 15.* Top 4 SPEC-identified clusters comparing CLIP and DINOv2 embeddings on ImageNet-1k dog breeds. The last row shows the UMAP representation of the top 10 SPEC-identified clusters for each embedding.

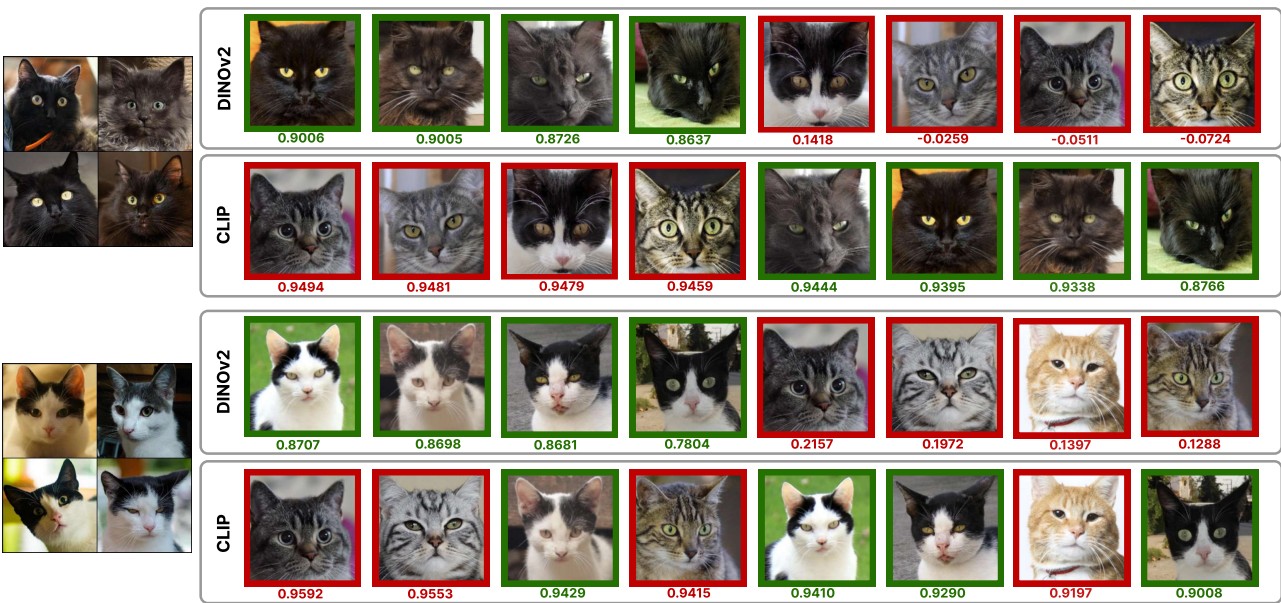

*Figure 16.* Comparing similarity ranking for SPEC clusters in DINOv2-CLIP on the AFHQ dataset. The leftmost images show the top 4 samples of two SPEC-identified clusters. Cosine similarity is computed with 4 cluster members (green-bordered) and 4 random images (red-bordered), sorted by score. Unlike DINOv2, CLIP ranks some random samples higher than cluster members.

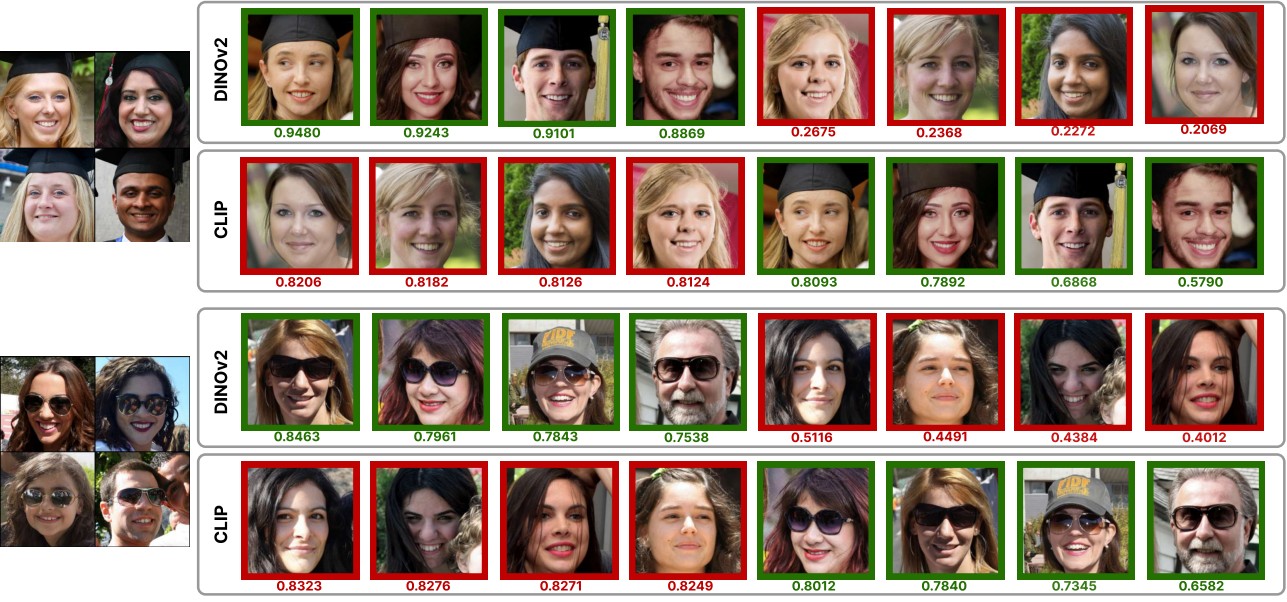

*Figure 17.* Comparing similarity ranking for SPEC clusters in DINOv2-CLIP on the FFHQ dataset. The leftmost images show the top 4 samples of two SPEC-identified clusters. Cosine similarity is computed with 4 cluster members (green-bordered) and 4 random images (red-bordered), sorted by score. Unlike DINOv2, CLIP ranks some random samples higher than cluster members.

## Comparing Different Image Embeddings using SPEC on MS-COCO dataset

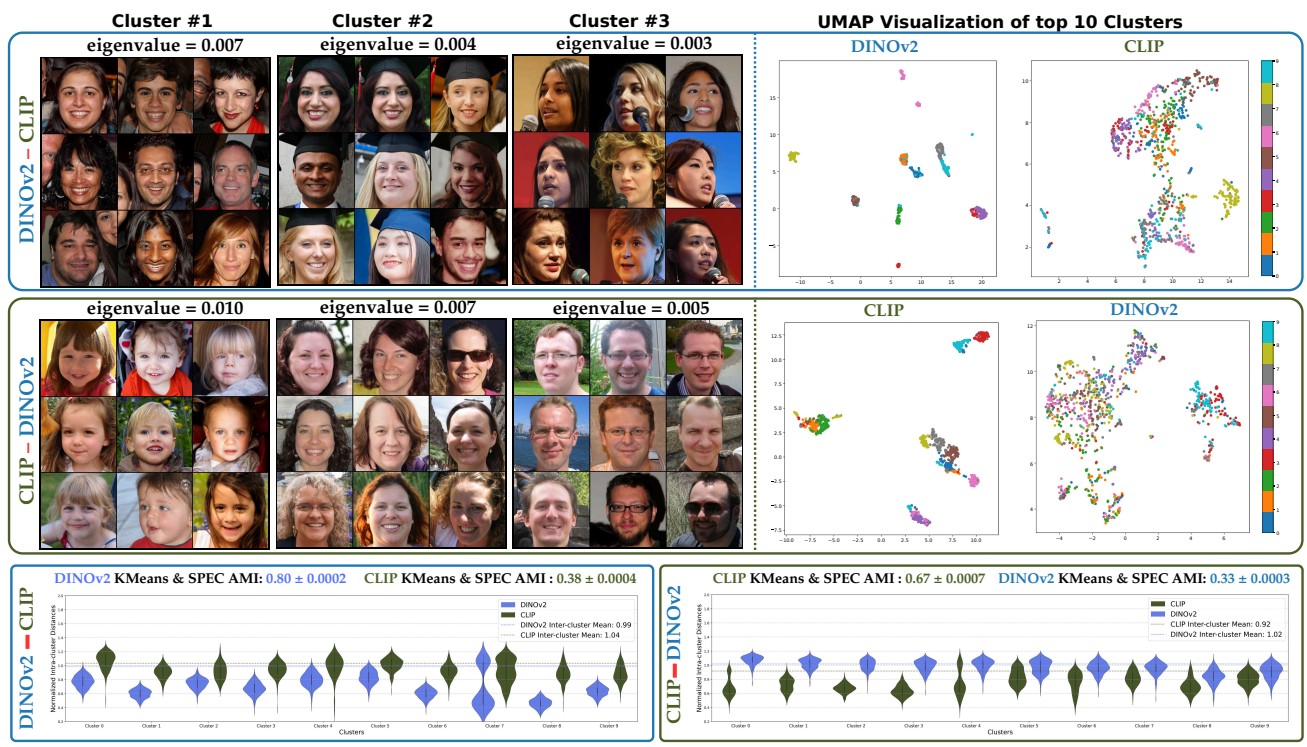

*Figure 18.* Comparing Different embeddings on the 120K samples from MS-COCO 2017 dataset.

*Figure 19.* Comparing embeddings on 70K FFHQ samples. Top numbers show SPEC cluster eigenvalues. Last two images per row display UMAP representations of SPEC clusters for each embedding.

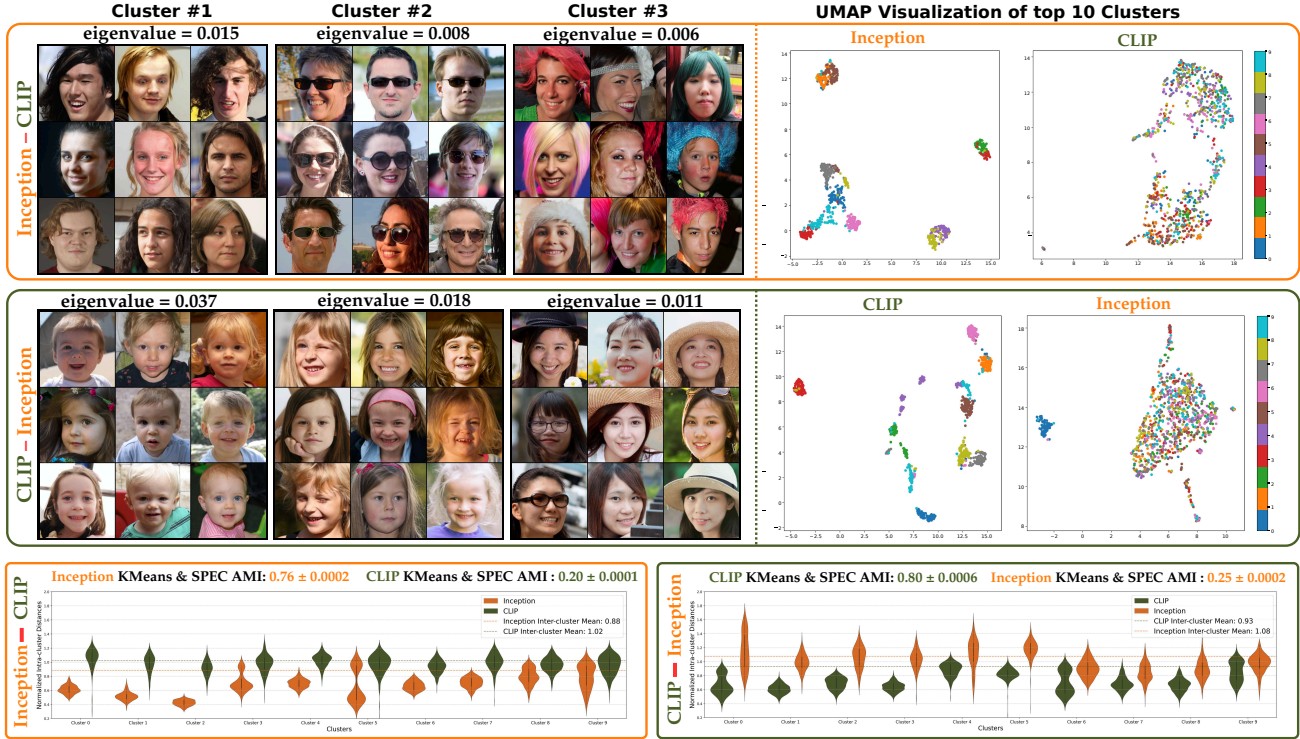

*Figure 20.* Comparing embeddings on 70K FFHQ samples. Top numbers show SPEC cluster eigenvalues. Last two images per row display UMAP representations of SPEC clusters for each embedding.

### B.6. SPEC-align Experiments

**SPEC-align using $\ell_2$-Norm of Kernel Difference Matrix.** As mentioned in Remark 2, another choice of SPEC-diff uses the $\ell_2$-norm of the eigenvalues of the kernel difference matrix $\Lambda_{\psi_1,\psi_2}$. Similar to the experiment in Figure 5, we aligned CLIP to DINOv2 using this $\ell_2$-norm-based objective. Figure 22 shows the CLIP kernel before and after alignment, alongside the target DINOv2 kernel. We also plot the SPEC-diff score over iterations, demonstrating successful convergence of the alignment process.

**SPEC-align Finetuning Experiments Parameters.** The following parameters were used in our experiment. We used the OpenCLIP GitHub repository (link) and used the MS-COCO 2017 training set, which consists of 120K pairs of texts and images. We use SPEC-align with the following parameters and chose DINOv2-Vit-B/14 as our reference model.

**Comparison of Kernel matrices for SPEC-align.** As shown in Figure 26, the SPEC-aligned CLIP kernel captures the top four clusters based on image content rather than the overlaid text labels. Furthermore, according to the t-SNE of the models, the SPEC-align cluster of fish is close to the cluster of images with overlaying text of fish showing that SPEC-align captured the similarity of text and image in this experiment while clustered based on the ground truth (cluster of images).

**Aligning text embeddings.** In addition, we aligned CLIP text features to the T5-XL model. In Figure 25, we can observe that the CLIP kernel has become more similar to T5-XL, and the SPEC-diff is also decreasing.

**Clusters Comparison of SPEC-align.** We provide additional results by comparing the top 8 Kernel-PCA (Gaussian RBF kernel) clusters of CLIP, DINOv2, and SPEC-align CLIP. We used the CLIP aligned with DINOv2 on the ImageNet training set. We compare the clustering of these embeddings with ImageWoof and the text-overlaid dataset in Figure 3. In Figure 23, we observe the top 8 clusters on the text-overlaid dataset. DINOv2 clusters are based on the images, while CLIP clusters are based on images and texts and in some cases fail to cluster based on the image. On the other hand, SPEC-align CLIP clusters based on images while focusing on the images with the same text, as expected. The top 8 clusters of ImageWoof in Figure 24 also show that CLIP clusters the dogs based on the gesture or their interaction with humans or the number of dogs, while DINOv2 clusters them only based on their breed. But SPEC-align CLIP clusters dogs based on their breeds while

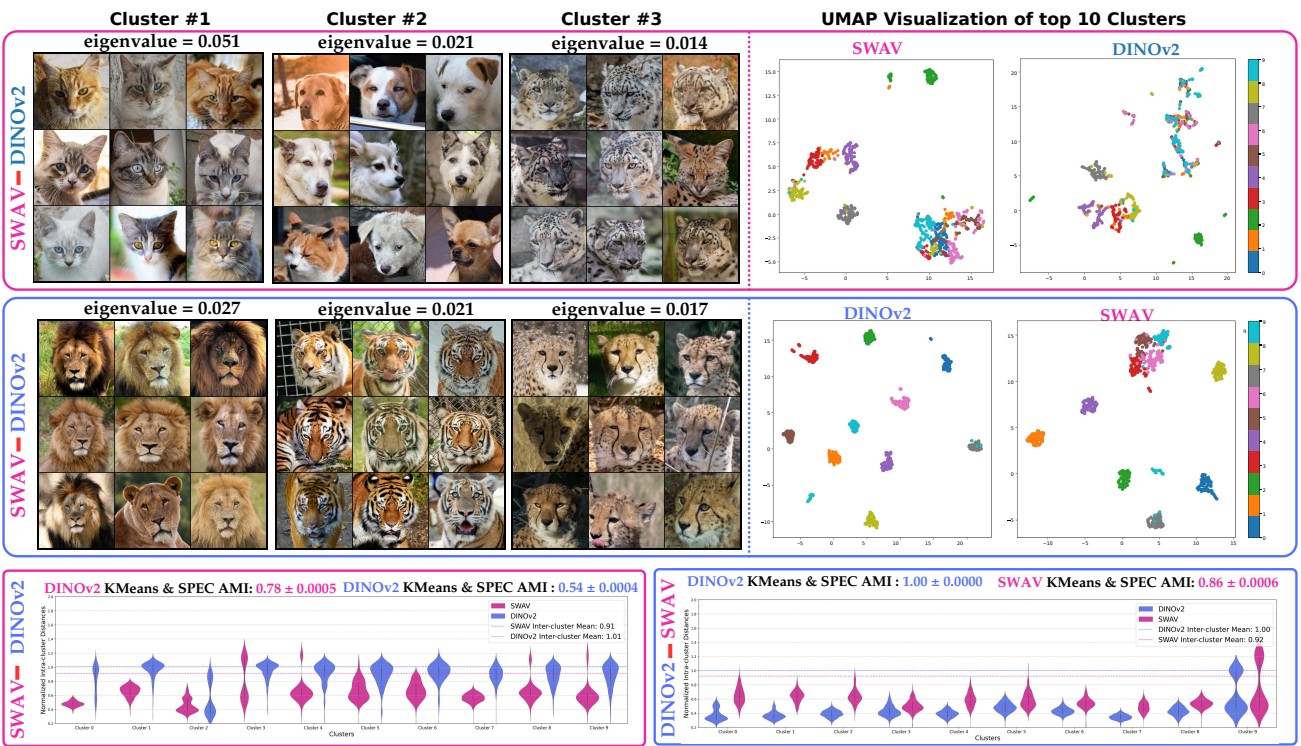

*Figure 21.* Comparison of different embeddings on 15K samples from the AFHQ dataset, consisting of 5K cats, 5K wildlife, and 5K dogs. The number at the top of each image represents the eigenvalue of the corresponding SPEC cluster. The last two images in each row show the UMAP representation of the SPEC clusters for each embedding individually.

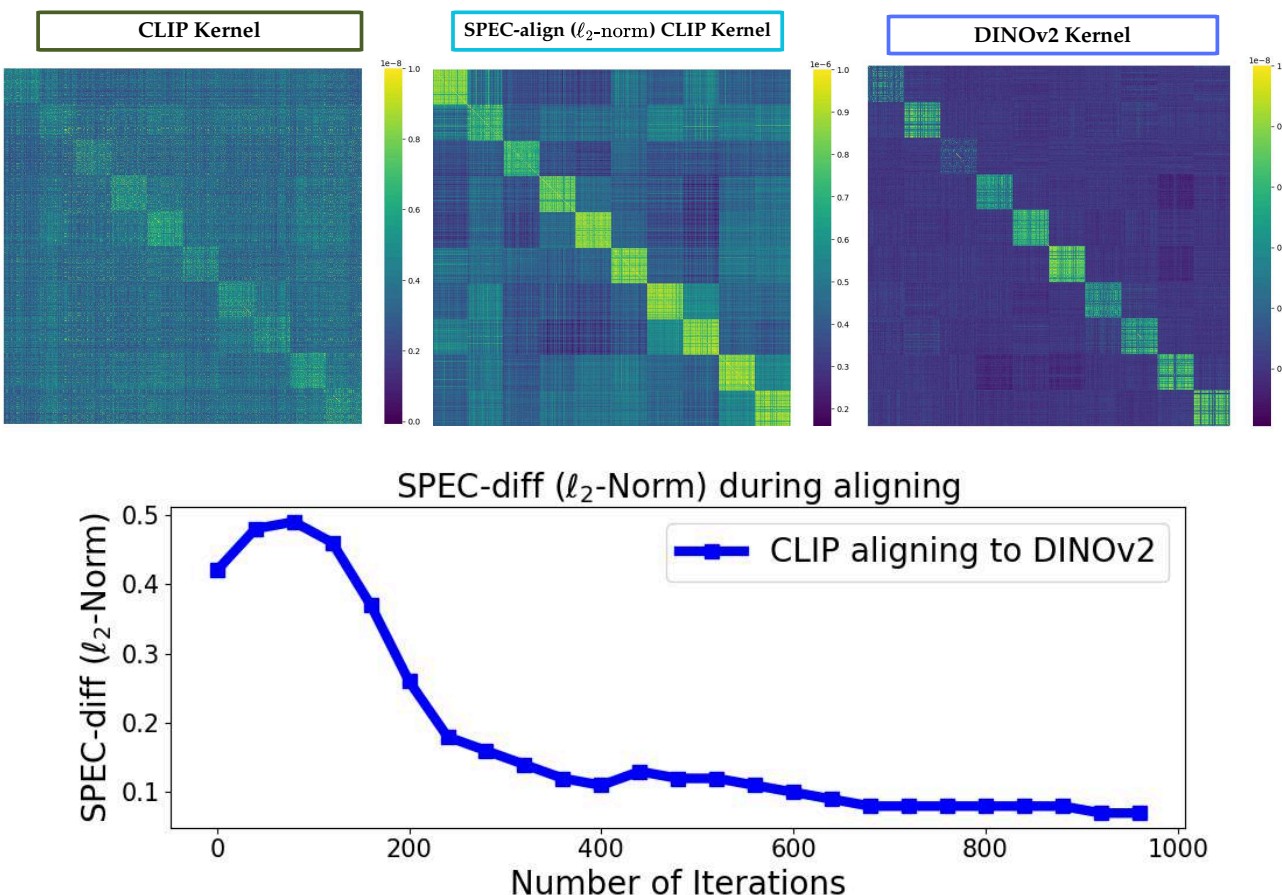

*Figure 22.* SPEC-align using the $\ell_2$-norm of the eigenvalues of the kernel difference matrix. Left: original CLIP kernel. Center: aligned CLIP kernel using $\ell_2$-norm-based SPEC-align. Right: target DINOv2 kernel. Bottom: SPEC-diff score ($\ell_2$-Norm) over alignment iterations.

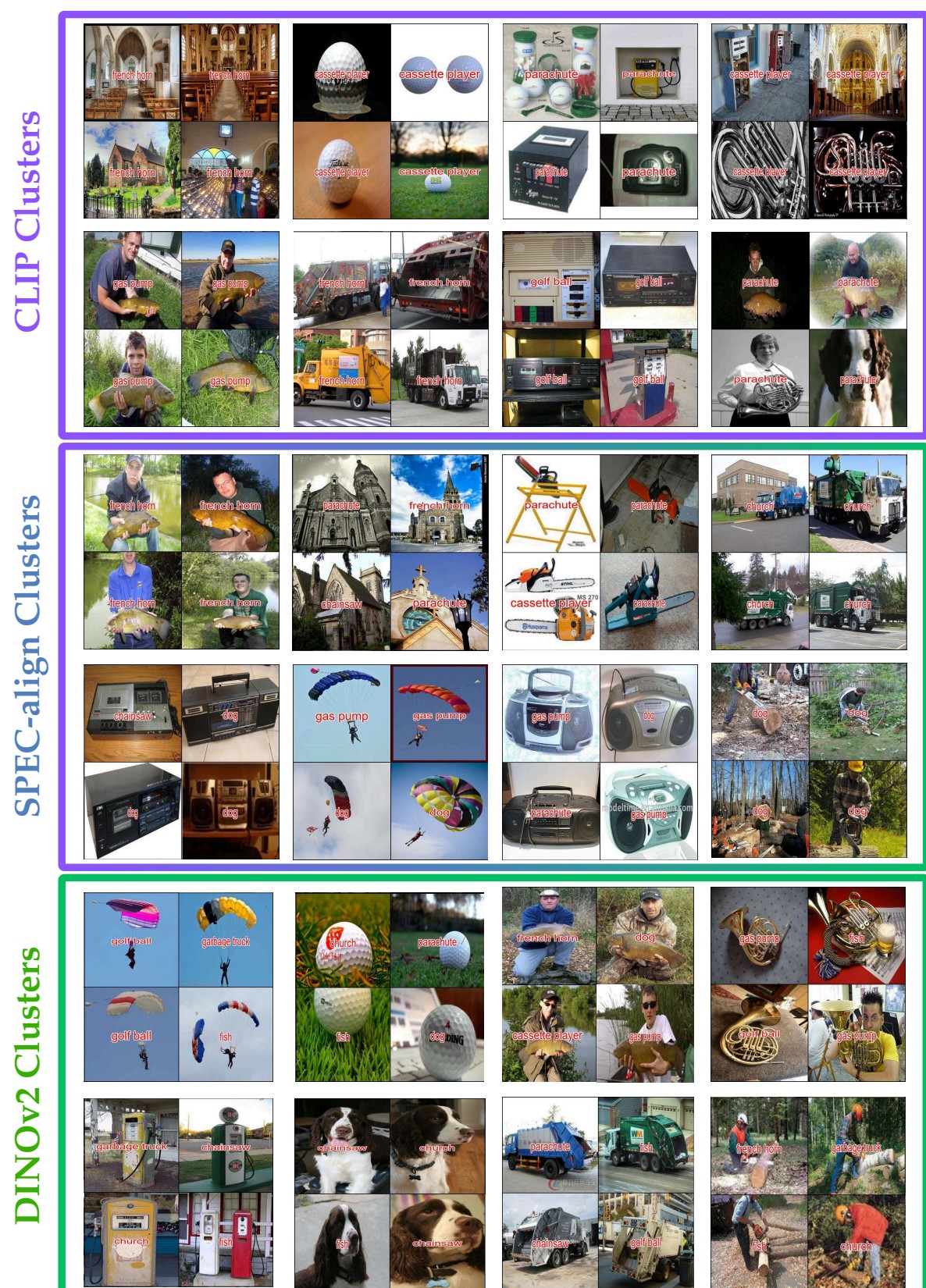

*Figure 23.* Top 8 Kernel-PCA (Gaussian RBF kernel) clusters for CLIP, DINOv2, and CLIP aligned with DINOv2, trained on the ImageNet dataset.

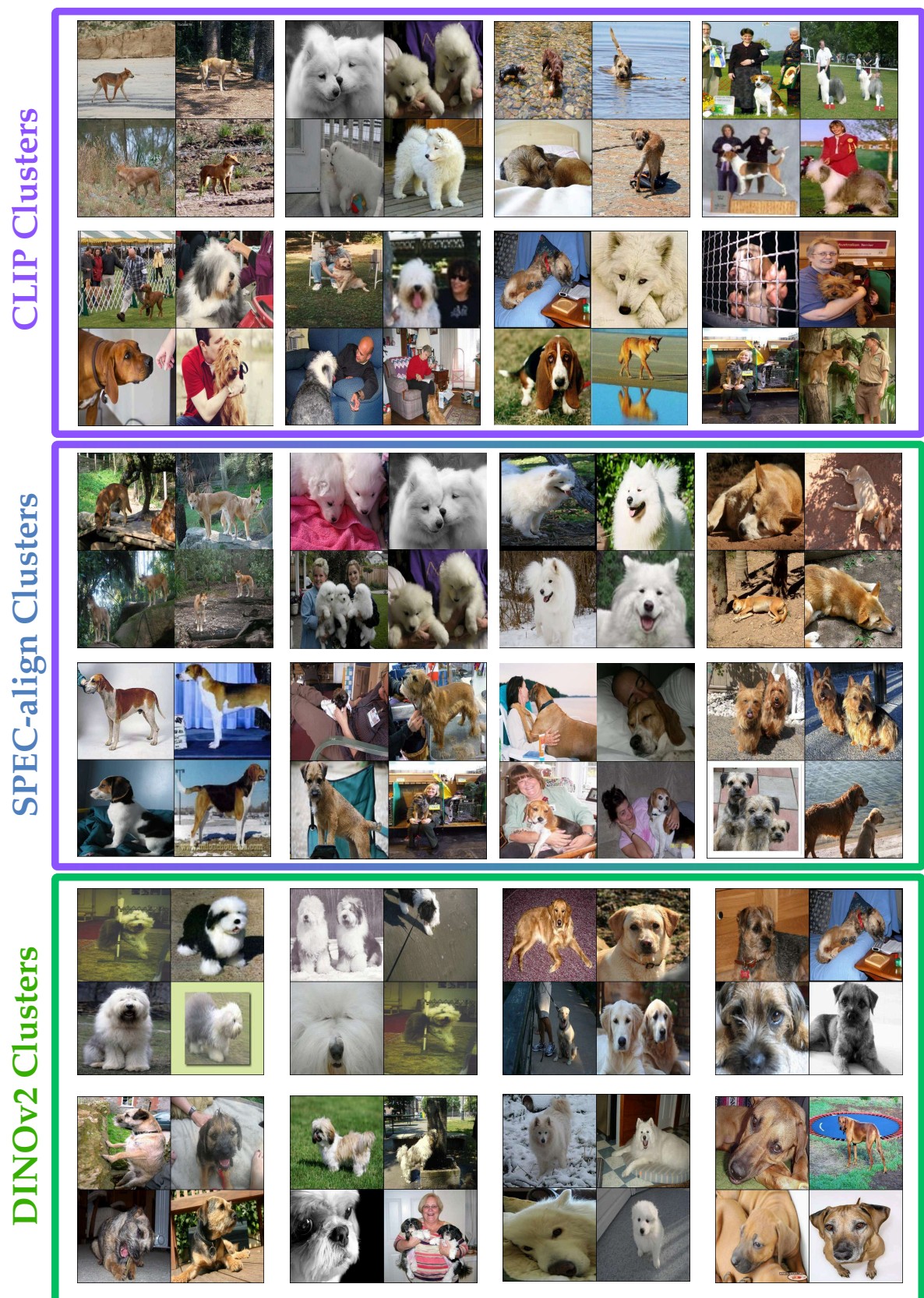

*Figure 24.* Top 8 Kernel-PCA (Gaussian RBF kernel) clusters for CLIP, DINOv2, and CLIP aligned with DINOv2, trained on the ImageNet dataset.

| Parameter | Value |
|---|---|
| accum_freq | 1 |
| alignment_loss_weight | 0.1 |
| batch_size | 128 |
| clip_alignment_contrastive_loss_weight | 0.9 |
| coca_contrastive_loss_weight | 1.0 |
| distributed | True |
| epochs | 10 |
| lr | 1e-05 |
| lr_scheduler | cosine |
| model | ViT-B-32 |
| name | Vit-B-32_laion2b_e16_freeze_5 |
| precision | amp |
| pretrained | laion2b_e16 |
| seed | 0 |
| wd | 0.2 |

*Table 2.* Configuration parameters used in the experiments.

*Table 3.* Linear evaluation of frozen features on fine-grained benchmarks.

| Model | Architecture | Data | Imagenet-1K |
|---|---|---|---|
| OpenCLIP | ViT-B/32 | LAION 400M | 73.50 |
| SPEC-align OpenCLIP | ViT-B/32 | LAION 400M | 76.45 |
| DINOv2 | Vit-B/14 | LVD-142M | 78.99 |

focusing on the gesture or the number of dogs or their interactions with humans.

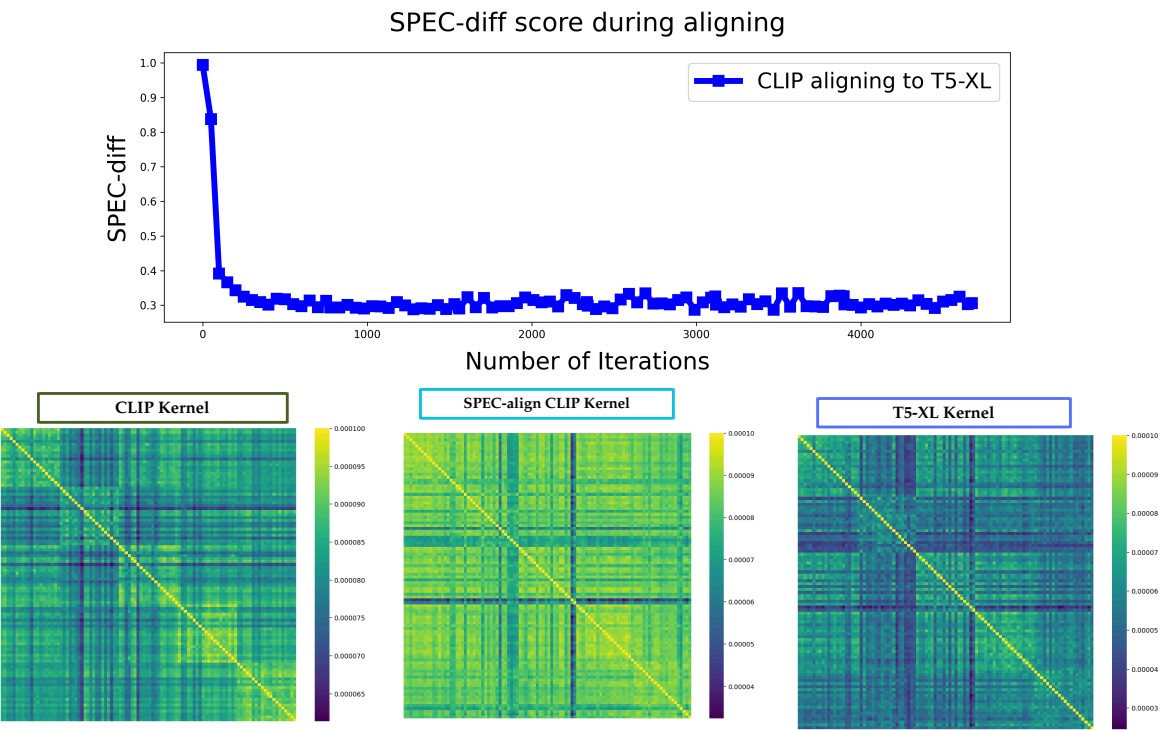

*Figure 25.* Comparison of Kernel matrices after using SPEC-align to match the sample clusters of CLIP to T5-XL with measuring SPEC-diff during the training.

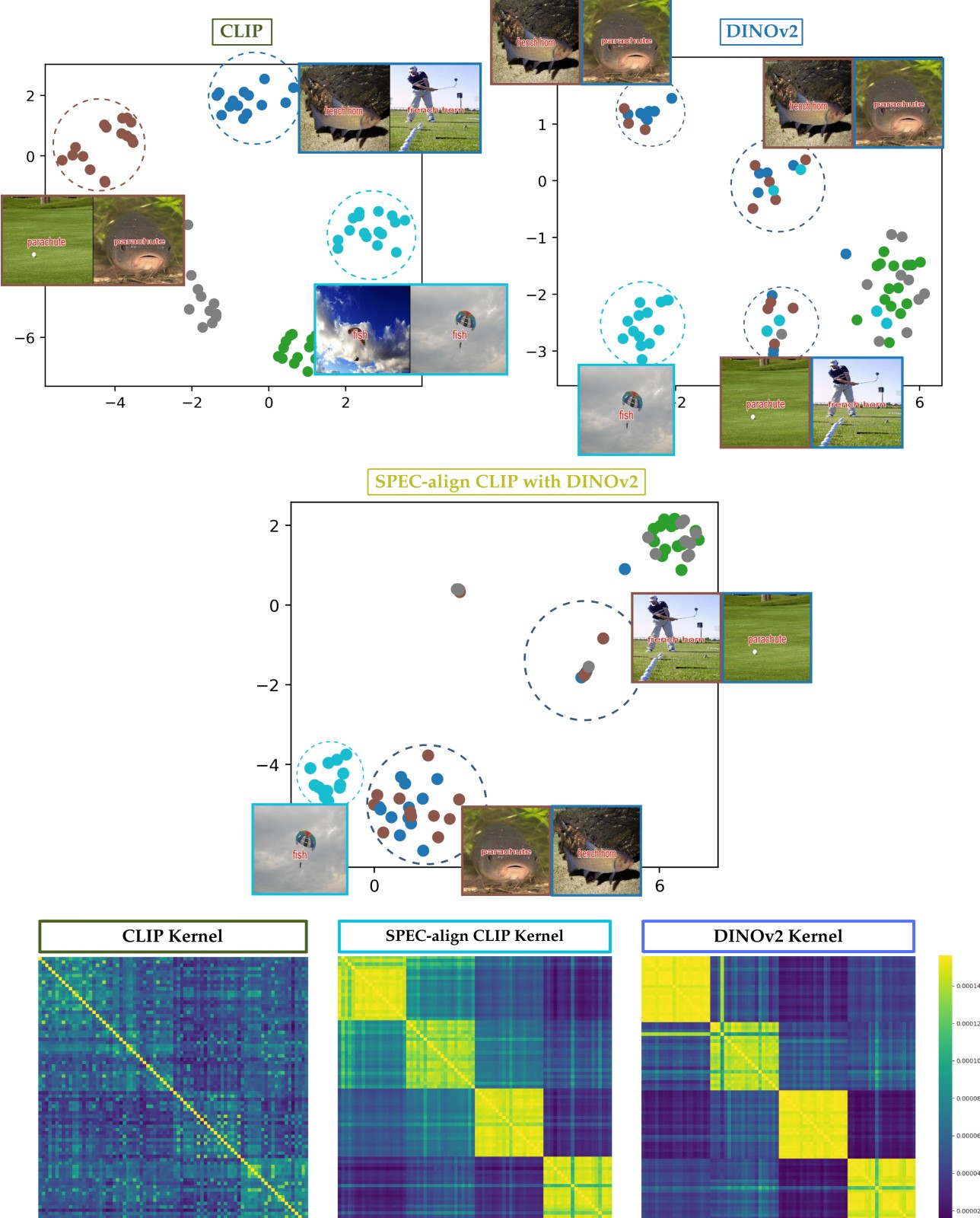

*Figure 26.* Comparison of Kernel matrices after using SPEC-align to align CLIP to DINOv2.

