# OpenReview forum: "Towards an Explainable Comparison and Alignment of Feature Embeddings"
_ICML.cc/2025/Conference — ICML 2025 poster_

### Official Review · Reviewer_KUAe · 2025-03-13

**Overall Recommendation:** 4

**Summary:**

The authors propose the Spectral Pairwise Embedding Comparison (SPEC) framework for comparing feature embeddings in an explainable manner. The goal is to identify differences in how two embeddings cluster data points, rather than relying solely on downstream performance metrics. The main contributions are: 1) a spectral analysis approach leveraging eigendecomposition of differential kernel matrices to detect mismatches in clustering behavior, 2) a scalable implementation that reduces computational complexity, 3) the SPEC-align method for aligning embeddings by minimizing clustering differences, and 4) numerical results demonstrating the effectiveness of SPEC on benchmark datasets such as ImageNet and MS-COCO. The study shows that SPEC can reveal embedding discrepancies and improve cross-modality alignment, such as enhancing CLIP embeddings with single-modality features.

**Claims And Evidence:**

The authors claim that the SPEC framework enables explainable comparisons of feature embeddings by identifying clustering differences between them. They support this claim through theoretical derivations using spectral analysis. They show that the eigendecomposition of the differential kernel matrix highlights mismatches in clustering behavior. The paper also introduces SPEC-align, which aligns embeddings by minimizing these differences.

**Essential References Not Discussed:**

Most of the essential references were cited in the paper.

**Experimental Designs Or Analyses:**

1. The authors evaluated the SPEC framework using well-known benchmark datasets, including ImageNet and MS-COCO.
2. The authors compared different feature embeddings using the eigendecomposition of differential kernel matrices to identify differences in clustering behavior.
3. The scalability of the approach is validated by implementing an optimized computation strategy for large datasets.

**Methods And Evaluation Criteria:**

The proposed methods and evaluation criteria are well-suited for the problem of explainable embedding comparison. Furthermore, the evaluation is robust. The authors used well-established benchmark datasets such as ImageNet and MS-COCO. The scalability of SPEC is validated through efficient computation techniques, making it feasible for large datasets.

**Other Comments Or Suggestions:**

1. Overall the paper is well-written.
2.  "eigendecomposin" → "eigendecomposition"
3. consistently use "difference kernel matrix" or "differential kernel matrix" to avoid confusion. Can you check this?

**Other Strengths And Weaknesses:**

Strengths:

1. The paper presents a novel spectral framework (SPEC) for comparing embeddings.
2. The SPEC-align method provides a structured way to align embeddings, resulting in improved cross-modality performance.
3. The scalability of the approach is well-addressed. This makes it feasible for large datasets.
4. The authors demonstrated the effectiveness of SPEC in detecting clustering differences across various embeddings and experiments.

Weaknesses:

The paper could benefit from direct comparison with existing embedding alignment techniques.

**Questions For Authors:**

1. The paper uses both "difference kernel matrix" and "differential kernel matrix." Are these terms interchangeable, or do they refer to distinct concepts?
2. The SPEC framework focuses on spectral analysis for comparing embeddings. How would it perform against non-spectral clustering methods such as hierarchical clustering or DBSCAN?
3. The paper discusses the computational efficiency of SPEC, but how does SPEC-align scale in high-dimensional spaces with very large datasets?
4. The paper introduces SPEC-align for embedding alignment, but how does it compare to existing alignment techniques such as Procrustes analysis or Wasserstein distance-based alignment?

**Relation To Broader Scientific Literature:**

The SPEC framework proposed by the authors leverages the eigendecomposition of differential kernel matrices to compare embeddings. This aligns with spectral clustering and diffusion maps, which are widely used in manifold learning within the existing scientific literature.

**Theoretical Claims:**

The theoretical claims presented in the paper appear well-founded. Specifically, the authors provide rigorous proofs demonstrating that the eigendecomposition of the differential kernel matrix effectively identifies clustering differences between embeddings. Moreover, they establish the scalability of their method by proving that the computational complexity grows linearly with the sample size.

---

> ### Author Rebuttal · Authors · 2025-04-01
>
> We thank Reviewer KUAe for the thoughtful and constructive feedback on our work. Below is our response to the comments and questions in the review: ([Our Numerical results are shown in this link](https://github.com/ICML6204/ICML6204/blob/main/ICML_Rebuttal.pdf))
>
> **1- Comparison with existing embedding alignment techniques**
>
> A distinction of our proposed alignment approach is that SPEC-Align operates directly on kernel matrices, enabling cluster-level comparison between encoders without requiring pointwise embedding alignment. The standard Procrustes and Wasserstein distance-based alignments usually enforce alignment in the sample level, which is a stronger alignment requirement compared to SPEC-align. This cluster-centric perspective is particularly advantageous when the embeddings have different topologies that should not be forcefully aligned or when only cluster-assignment consistency (rather than pointwise matching) is needed for the downstream task. We will include this discussion in the revised text.
>
> **2- Difference kernel matrix and differential kernel matrix**
>
> We thank the reviewer for pointing this out. We will update the paper and consistently use the term “difference kernel matrix” to avoid any confusion.
>
> **3- Non-spectral clustering methods such as hierarchical clustering or DBSCAN**
>
> While the current SPEC framework follows spectral clustering for kernel-based embedding comparison, extending non-spectral clustering methods such as hierarchical clustering or DBSCAN [1] to perform cluster-based embedding comparison will be an interesting direction for future exploration. We will discuss this future direction in the conclusion section.
>
> [1] Ester et al., "A density-based algorithm for discovering clusters in large spatial databases with noise" KDD 1996
>
> **4- SPEC-align computational complexity and scalability in sample size**
>
> Following Proposition 5.1, we showed that SPEC-align can be computed using the power method on a matrix with size $(d_1+d_2)\times (d_1+d_2)$, and the gradient calculation can be run with only $O(\max\lbrace n_B, (d_1+d_2)^2\rbrace)$ computations ($O((d_1+d_2)^2)$ is the cost of one run of the power method), considering a batch size $n_B$ and embedding dimensions $d_1$ and $d_2$. We will explain the complexity of SPEC-align in the revised text.

---

### Official Review · Reviewer_7vC9 · 2025-03-14

**Overall Recommendation:** 4

**Summary:**

The work is proposed to allow an interpretable comparison of feature embeddings from different methods through a Spectral Pairwise Embedding Comparison (SPEC) on clustering and also proposes an approach to align one embedding with another. The work is relevant for a wide audience where interpretability and flexibility with embeddings is desired.

**Claims And Evidence:**

The contributions are centered around the proposed SPEC method for comparing two embeddings, an O(n) implementation for the same along with an alignment method to align one embedding with another. Evidence has been provided using methods, experiments and insights which are discussed in other comments.

**Essential References Not Discussed:**

I am not aware of any essential references that have not been discussed.

**Experimental Designs Or Analyses:**

- Experiments for embedding comparisons have been performed in Section 6 (and supplementary material) where the experiment settings have been reported. Figure 1 shows the top-3 clusters based on the approach1 (DINOv2, CLIP, SWAV) in comparison to approach 2 (CLIP, DINOv2, DINOv2) which is validated through the tSNE representation of each embedding individually. The pairwise SPEC embedding approach provides insights like 'CLIP ranking random samples higher as compared to DINOv2' which is qualitatively shown in Figure 8,9. Experiments have also been performed by performing a typography attack to highlight that CLIP clusters are based on overlaid text whereas DINOv2 focuses on overlaid text.
- The alignment of CLIP with DINOv2 to improve CLIP's performance on MS-COCO 2017 is highlighted in figure 3, 12 where the alignment of Kernel matrices of CLIP to DINOv2 is performed to obtain the SPEC-align CLIP Kernel using the penalty based alignment objective discussed in the methods.

**Methods And Evaluation Criteria:**

- The study uses four widely used benchmark image based datasets, a dataset constructed by overlaying text labels on the ImageNet-1k dataset and text based datasets generated by GPT-4o.
- Embeddings used for the evaluation are fairly representative of the commonly used embedding methods (CLIP, DINOv2, etc.)
- The methods are described in detail -- the algorithm has been described for spectral pairwise embedding comparison (SPEC) using the kernel covariance matrices and the loss for aligning embedding maps of one approach with another by adding a penalty term based on the SPEC-diff term that penalizes the mismatch between the with the reference embedding.

**Other Comments Or Suggestions:**

- It would be good to provide the API endpoints for the embeddings used in this work. The methods have been cited but the endpoints would help in reproducibility of the approach.
- As there have been several insights provided by comparing embedding methods like CLIP and DINOv2 by highlighting cases where the embeddings have some limitations. It would be good to provide some general guidelines about how to use the SPEC approach for the broader audience which may be interesting in comparing their embeddings on general tasks. For instance, the insight provided in Subsection "SPEC comparison of embeddings on different image and text datasets." regarding the RoBERTa clustering based on gender and profession could have fairness implications for the wider audience. It would be good to provide a detailed implementation for further validation of the approach in other domains (and datasets) and ensure reproducibility of the results.
- In Section B.4.1, does the 'alignment_loss_weight' parameter refer to the β term in Eqn: 6? It may be good to clarify the difference between 'clip_contrastive_alignment_loss_weight' and 'alignment_loss_weight'. I can see that the 'coca_contrastive_loss_weight' is mentioned in the OpenCLIP github repository but the alignment terms may need some clarification.
- In the last line of B.1,  'was' can be removed from '..observe that E5 was managed to cluster captions...'
- For the alignment approach, there could be some explanation on the benefits of alignments other than CLIP->DINOv2. For instance, it would be good to see some experiments on text embedding alignment and/or to see the behavior of embeddings upon performing other types of alignment. It would also be good to discuss how the role of β penalty on the alignment.

**Other Strengths And Weaknesses:**

The paper is well-written and readable. References have been provided to point to the existing literature where needed. The framework of embeddings comparison and alignment is defined well and has been theoretically and experimentally validated extensively.

**Questions For Authors:**

I have mentioned my comments in other sections.

**Relation To Broader Scientific Literature:**

Text and Image embeddings are widely used in many applications to perform downstream tasks or to just store objects for retrieval purposes. The interpretability of these embeddings is sought for in multiple areas like healthcare where situations like domain shift or bias in embeddings are undesirable. The pairwise embeddings comparison approach could allow the users to take a well informed decision for choosing the embeddings for the specific application and embedding-alignment approach could be beneficial in adapting embeddings for obtaining specific outcomes. This work is (probably) a first step towards providing insights on the commonly used embedding methods through experiments on the selected datasets.

**Theoretical Claims:**

I have not validated the proofs for the theoretical claims. However, the framework definition, formulation and proofs are consistent.

---

> ### Author Rebuttal · Authors · 2025-04-01
>
> We thank Reviewer 7vC9 for the thoughtful and constructive feedback on our work. Below is our response to the comments and questions in the review: ([Our Numerical results are shown in this link](https://github.com/ICML6204/ICML6204/blob/main/ICML_Rebuttal.pdf))
>
> **1- API endpoints for the embeddings used in the work**
>
> We would like to clarify that we have used open-source embeddings in our numerical experiments from their main repositories. Specifically, the embeddings discussed in the paper are in the following repositories, for which we will include the links in the revised paper:
>
> DINOv2 downloaded from: https://huggingface.co/docs/transformers/en/model_doc/dinov2
>
> CLIP downloaded from: https://huggingface.co/docs/transformers/en/model_doc/clip
>
> SWAV downloaded from: https://huggingface.co/lixiangchun/imagenet-swav-resnet50w2
>
> RoBERTa downloaded from: https://huggingface.co/docs/transformers/en/model_doc/roberta
>
> E5 downloaded from: https://huggingface.co/intfloat/e5-base-v2
>
> Inception V3 downloaded from: https://huggingface.co/docs/timm/en/models/inception-v3
>
>
> **2- alignment_loss_weight parameter in Section B.4.1**
>
> We thank the reviewer for pointing this out. The alignment_loss_weight parameter refers to the $\beta$ hyperparameter in Equation 6, which we have set as $0.1$ (mentioned in the Appendix) in addition to the parameters of the OpenCLIP Github repository. We will make this clear in the revision.
>
> **3- Additional experiments for text embeddings’ alignment**
>
> We thank the reviewer for the suggestion. To further address the reviewer's comment, we aligned the CLIP text embedding with the T5-XL model. In Figure 6 of the rebuttal link (last page), we can observe that the CLIP kernel has become more similar to T5-XL, and the SPEC-diff is also decreasing.
>
> T5-XL model huggingface: https://huggingface.co/google/t5-v1_1-xl
>
> **4- role of $\beta$ penalty on the alignment**
>
> In our experiments, we conducted a grid search for $\beta$ values ranging from 0.05 to 1. During our experiments, we observed that when $\beta$ is large ($\beta$ > 0.5), the consistency between CLIP text and image alignment decreases, and aligning should occur gradually. For our experiment, we ultimately used $\beta$ = 0.1.

---

> > ### Comment · Reviewer_7vC9 · 2025-04-06
> >
> > I thank the authors for addressing my concerns in detail. I believe that the work is valuable to the community and should be accepted.

---

### Official Review · Reviewer_iKx5 · 2025-03-23

**Overall Recommendation:** 4

**Summary:**

This paper proposes a method for comparing the embedding spaces for pairs of models by constructing PSD kernel matrices of each embedding space and studying the differences of these PSD matrices. The theoretical section discusses how, under some assumptions, the eigendecomposition of this difference between the PSD matrices carries the information about which clusters differ between the embeddings. The authors then perform an experimental analysis of their results.

**Claims And Evidence:**

The claims in the paper are supported by evidence, although the evidence could be made stronger. It seems that the author's primary mechanism for verifying their method is via tSNE plots of those points which are in line with the principal eigenvectors of the difference matrix. The idea here is that if these points are in tight clusters in the tSNE plot, then they must have a similarly nice structure in the embedding space. Although I agree with this argument in principle, I have two concerns with it. First, this is a qualitative measure and therefore cannot tell the full story. For example, Figure 1's third tSNE plot is not very clear. Which side is supposed to be better clustered?
My second concern is a more theoretic one about relying on tSNE outputs. As far as I know, given a tSNE output, it's not clear what can be confidently said about the corresponding input. Thus, if the tSNE output is clusterable, this does not necessarily guarantee that the input has a similarly nice structure. In short, the tSNE plots are not particularly convincing and, even if they were, I'm not sure what information can be gleamed from them.

Similarly, I am not sure that Figures 8 and 9 are the best format in which to make the point the authors are making. I agree that if the center of a cluster's nearest neighbors belong to tha same cluster, then the cluster potentially has a nicer structure than one whose centroid has nearest neighbors from other clusters. However, I again have two concerns with this being a definitive statement. First, it could certainly be the case that it is purely by luck that this structure emerges. For example, suppose there is a cluster, $C_1$, which is perfectly separable from the other ones except for two points which land within it from another cluster $C_2$. Then $C_1$ is precisely the kind of cluster the authors are looking for. However, these two points which belong to $C_2$ could be in the center of $C_1$, thereby skewing the author's nearest-neighbors-to-the-center metric. (I also have slight concerns about figures showing results being relegated to the appendix but discussed in the main body of the paper, but it's not a big deal).

Rather than the metrics the authors utilized, I would instead be much more interested in metrics which are (a) quantitative and (b) global. For example, one could fit k-means to the clusters found by SPEC and evaluate the cost of the k-means clusters. Additionally, one could visualize the variance of the embedding space by showing violin plots of the distances of points in the clusters to the center. These kinds of plots would unambiguously make the authors' point by showing that SPEC finds clusters which are compact and separable in one embedding space but have poor structure in the other one. As it stands now, this point (which is the premise of the paper) has not been shown unambiguously by the results.

**Essential References Not Discussed:**

All of the essential references seem present. I would, however, note that the way the references are included is slightly strange. There are 0 references on the first page and then a littany of them in the related work section. It would seem appropriate to include some references in Sections 1, 3 and 4 to back up the statements which are made. Here's an example sentence which would do well to have a few references to support it: "Understanding these differences can aid in interpreting and debugging embeddings and can also be leveraged to align multiple embeddings.  Furthermore, interpreting the discrepancies between embeddings can be utilized to select representation models for downstream applications such as generative model evaluation."

**Experimental Designs Or Analyses:**

My thoughts on the experimental design and analysis are outlined above.

**Methods And Evaluation Criteria:**

The datasets and models used in the paper are reasonable for proving the authors' point.

**Other Comments Or Suggestions:**

No other comments come to mind.

**Other Strengths And Weaknesses:**

A strength of this paper that I have not focused on enough in the review is its clear description and intuitive idea. I am surprised this has not been done before and it feels like a very natural way to establish the differences between learned representations. As soon as it was described in the intro, I understood the point and it was obvious to me that it should work reasonably well.

One weakness which I feel needs to be mentioned is that I am not convinced that this actually aids in explainability. Certainly, the method makes it clear which samples one model has clustered better than another model, but couldn't this be evaluated using the classification accuracy on those samples? Why is this method necessary in a way that distinguishes it from other standard explainability measures?

**Questions For Authors:**

The questions for the authors can be found scattered throughout the review. Perhaps the most pressing one is in finding quantitative metrics and analysis which support the claims. I would be interested in seeing analyses of the distances from points to the cluster centers across the sets of points identified by SPEC. The absence of quantitative supporting evidence leaves the paper less convincing than it otherwise would be.

The second question that would be good to address would be how this differs in a measurable way from other explainability measures which compare between models. What does SPEC tell the user which is complementary to, for example, evaluating the classification accuracy of various samples between models? This would again require quantitative evidence to back it up.

**Relation To Broader Scientific Literature:**

I am not well-acquainted with the broader scientific literature regarding explainability via studying the clusterability of the embedding spaces. However, this work is clearly in line with the themes from kernel-based ML literature. I appreciate the author's reference to Kernel PCA, as they are essentially doing PCA using the difference matrix rather than the kernel matrices themselves. Perhaps some references to explainability via principal components of the embedding distribution would be nice to add.

**Theoretical Claims:**

The theoretical claims are reasonable and follow with what I would expect, having worked in kernel-based ML for a bit of time now. However, I would be curious whether the authors can evidence experimentally that their assumptions (Conditions 1 and 2) are reasonable to expect in practice. Specifically, what are the bounds on $\varpesilon_1$ and $\varepsilon_2$ on the datasets the authors used in the paper? What is the value of $\zeta$ which the authors obtain and is the bound in Theorem 4.1 supported experimentally?

---

> ### Author Rebuttal · Authors · 2025-04-01
>
> We thank Reviewer iKx5 for the thoughtful and constructive feedback on our work. Below is our response to the comments and questions in the review: ([Our Numerical results are shown in this link](https://github.com/ICML6204/ICML6204/blob/main/ICML_Rebuttal.pdf))
>
> **1- Quantitative evaluation of the SPEC method**
> We thank the reviewer for the suggestions. Following the suggestion, we have analyzed the cluster distributions using violin plots to visualize normalized distances between data points. The plots also suggest that one embedding can cluster the points more strongly. Additionally, we performed KMeans clustering on the embedding features and computed normalized mutual information (NMI) between SPEC labels and KMeans clusters. The results indicate that one embedding reaches a stronger alignment and correlation with the KMeans labels.
>
> **2- Referring to the Appendix figures in the main text**
>
> We thank the reviewer for pointing this out. We will move the referrals to Figures 8, 9, and 12 to the Appendix in the revision.
>
> **3- Experimental Validation of Assumptions: Conditions 1 and 2**
>
> We analyzed Conditions 1 and 2 in the experiment of Figure 2 in the main text. The numerical results are presented in the Rebuttal Figures 1 and 2.
>
> **4- References in Sections 1 and 3**
>
> We thank the reviewer for pointing this out. We will cite references related to the spectral clustering, kernel PCA, alignment of embeddings, and explainability of PCA approach to embedding analysis in the revised sections 1 and 3.
>
> **5- SPEC method vs. using classification accuracy for embedding comparison**
>
> We appreciate the reviewer’s point on using the labeled samples by embedding models to explain their differences. We note that the selection of the label set can significantly influence the result of this approach. On the other hand, the SPEC method performs unsupervised learning to identify the soft-labels (i.e., eigenvectors) for performing the comparison of embeddings. Therefore, the SPEC method can avoid any biases that might be introduced by using a given label set. Also, we highlight that the SPEC method performs a soft clustering, which is different from a hard labeling of the samples assigning each data point to only one label. We will discuss these points in the revised introduction and conclusion.

---

> > ### Comment · Reviewer_iKx5 · 2025-04-02
> >
> > I thank the authors for their hard work.
> >
> > Disclaimer: I will not open any links as it would be unfair to authors of other papers I'm reviewing who stuck to the 5K character limit. The authors are welcome to describe their results in text if they wish.
> >
> > Regarding point (5) in the rebuttal, I'm still not convinced. Yes -- the clustering results are unsupervised and therefore cannot be biased. But... the labels DO correspond to the class distinctions, almost by definition. Even if it's not a perfect fit. So wouldn't the labels still give signal to which things were grouped together differently by the different models? The suggestion that one shouldn't use labels towards explainability seems contrived to support the authors' proposed method.
> >
> > In either case, I am keeping my score and believe this paper should be accepted. These are simply details.
> >
> > ------
> >
> > EDIT: Thank you for the comment! I didn't realize links were allowed.
> >
> > I looked through the outputs and they indeed look quite convincing. I feel that my concerns have been addressed appropriately. I appreciate the hard work.

---

> > > ### Author Response · Authors · 2025-04-02
> > >
> > > We sincerely thank Reviewer iKx5 for the thoughtful feedback on our response. We appreciate the reviewer’s point on the 5000 character limit for the response. We would like to point out that our provided anonymized link follows the rules in the conference website (https://icml.cc/Conferences/2025/PeerReviewFAQ). In the following, we provide a summary of the URL's related numerical results in text format:
> > >
> > > **Quantitative evaluation of the SPEC method (Violin plots)**
> > >
> > > Following the reviewer’s suggestion, we have analyzed the cluster distributions using violin plots to visualize normalized distances between data points. In the tables below, we report the mean and standard deviation of pairwise distances of embedded data in each cluster (normalized by the average pairwise distance over all the data pairs for each embedding to ensure a fair comparison between the embeddings).
> > >
> > > Table 1: The averaged pairwise distance of embedded data pairs in the clusters of Figure 2 in the main text.
> > >
> > > |                       | Inter-Clusters      | Cluster #1       | Cluster #2      | Cluster #3      | Cluster #4       | Cluster #5       |
> > > |-------------------------------------|-----------------|------------------|-----------------|-----------------|------------------|------------------|
> > > | **CLIP Clusters (CLIP - DINOv2)**   | 0.96 $\pm$ 0.01 | 0.82 $\pm$ 0.186 | 0.53 $\pm$ 0.09 | 0.52 $\pm$ 0.16 | 0.55 $\pm$ 0.12  |  0.57 $\pm$ 0.13 |
> > > | **DINOv2 Clusters (CLIP - DINOv2)** | 0.95 $\pm$ 0.02 | 1.00 $\pm$ 0.13  | 0.89 $\pm$ 0.13 | 0.85 $\pm$ 0.01 |  0.88 $\pm$ 0.02 | 0.92 $\pm$ 0.01  |
> > > | **CLIP Clusters (DINOv2 - CLIP)**   | 1.01 $\pm$ 0.09 | 0.95 $\pm$ 0.14  | 0.94 $\pm$ 0.09 | 0.89 $\pm$ 0.09 | 0.86 $\pm$ 0.12  | 0.85 $\pm$ 0.15  |
> > > | **DINOv2 Clusters (DINOv2 - CLIP)** | 1.02 $\pm$ 0.01 | 0.39 $\pm$ 0.06  | 0.37 $\pm$ 0.05 | 0.51 $\pm$ 0.04 | 0.42 $\pm$ 0.04  | 0.52 $\pm$ 0.07  |
> > >
> > > Additionally, we performed KMeans clustering on the embedded data vectors and computed normalized mutual information (NMI) between SPEC labels and KMeans-cluster labels. The results indicate that the source embedding reaches a stronger correlation with the KMeans labels.
> > >
> > > NMI results for Figure 1 of main text:
> > >
> > > | Embedding$_X$  - Embedding$_Y$ | K-means (Emb$_X$) & SPEC NMI |  K-means(Emb$_Y$) & SPEC NMI |
> > > |-------|-----------|------|
> > > | DINOv2 - CLIP             | 0.94 $\pm$ 0.0003              | 0.65 $\pm$ 0.0014              |
> > > | CLIP - DINOv2             | 0.78 $\pm$ 0.0006              | 0.44 $\pm$ 0.0002              |
> > > | SWAV - DINOv2             | 0.74 $\pm$ 0.0003              | 0.49 $\pm$ 0.0002              |
> > > NMI results for Figure 2 of main text:
> > > |                   | **CLIP** KMeans & SPEC NMI | **DINOv2** KMeans & SPEC NMI |
> > > |-------------------|----------------------------|------------------------------|
> > > | **DINOv2 - CLIP** | 0.33 $\pm$ 0.0016          | 0.96 $\pm$ 0.0005            |
> > > | **CLIP - DINOv2** | 0.68 $\pm$ 0.0005          | 0.16 $\pm$ 0.0001            |
> > >
> > >
> > > **Regarding Item 5 in our response**,
> > >
> > > We appreciate Reviewer iKx5's feedback on the response. We would like to clarify that we did not mean that the unsupervised SPEC method is universally better than a supervised comparison including the idea mentioned by Reviewer iKx5. In fact, we agree that in scenarios with available fine-grained labeled datasets, the supervised comparison can perform more efficiently compared to an unsupervised approach. On the other hand, datasets with sufficiently fine-grained labels are not always available for a general application. Also, even a seemingly fine-grained label set (like the labels of ImageNet) may not be comprehensive enough and lack some required details in a general case. In settings with unlabeled data or labeled samples lacking fine-grained labels, the unsupervised nature of SPEC can be beneficial. One future direction could be to propose a semi-supervised comparison approach that utilizes both the labeled and unlabeled samples. In the revision, we will include this discussion to better compare the unsupervised SPEC method vs. supervised and potential semi-supervised approaches that utilize labeled data.

---

### Official Review · Reviewer_hTAA · 2025-03-25

**Overall Recommendation:** 4

**Summary:**

The authors derive a method for identifying clusters which are strongly clustered by one encoder and weakly clustered by another encoder. The run time of the method scales linearly with the number of samples in the dataset. This is deployed on a few image datasets for some established image foundation models, and a collection of text samples for some established language foundation models. The authors also propose and briefly demonstrate that the metric can be deployed for fine-tuning to transfer the clustering of one encoder into another.

## update after rebuttal

I am happy to increment my score 3->4 after clarifications and requested changes from the authors during the rebuttal period.

*Note on Adjusted Mutual Information for new results*

It would make sense to either use NMI or AMI for all the results; it is redundant to show both, and inconsistent to swap between them. AMI is generally better than NMI because it accounts for the coincident information measurements due to chance. It's especially important when the number of clusters differs between clusterers to compare, hence that's when it becomes critical to use AMI instead of NMI. If you're in the situation where this sometimes happens, I recommend you swap to using AMI for *all* results.

**Claims And Evidence:**

Okay.

**Essential References Not Discussed:**

The relationship to Laplacian spectral clustering (e.g. Ng, et al, NeurIPS 2001) wasn't really discussed. I think it would be helpful to articulate the similarities and differences between the SPEC method and spectral clustering to the reader.

**Experimental Designs Or Analyses:**

Okay.

**Methods And Evaluation Criteria:**

The text dataset used is a bit opaque. It is not clear why GPT-4o was used to generate the text samples instead of using an existing text dataset, and it is also unclear what prompting was used to generate this text data.

It would be advantageous to include more quantitative evaluation. For example, does low SPEC-diff on a dataset imply that two models have similar performance at the task established by the labels for that dataset?

**Other Comments Or Suggestions:**

- L133 left. The matrix is positive semi-definite because k is an inner product, so $k(\cdot,\cdot)\in[0,1]$. However the more general statement for a kernel function only constrains its output to the reals. I think the implication could be made more clear by adding the missing step, e.g. on L131 by specifying $<\phi(x),\phi(x')>\in[0,1]$.

- L068 right. should be \citet not \citep if the citation is part of the text, as it is here. It is bad English grammar to write that something is in (a parenthetical description). As author names appear in the text, you should replace this with "has been studied by \citet{ref1,ref2,ref3,ref4,ref5}."

- L077 right. in this case the citation of Gao (2021) should be parenthetical, \citep, instead of textual. The Muennighoff citation is correctly textual \citet. However this sentence doesn't make sense anyway - how is Muennighoff (2023) offering something within the work of Gao (2021)?

- L080, L099, L103, L107, L369 citations should also be \citet

- L145 left. Bracket should go around the whole fraction (use `\left(` and `\right)`)

- L164 left. Should be $\phi_1$ and $\phi_2$, not $\phi_1$ twice.

- L180 left. Please hyperlink here to the proof within the Appendix. Similarly for other deferred proofs.

- L190 right. Missing subscript command on $\Gamma\Psi\Psi$.

- L191 right. if -> of

- L215 right. shift-invarian -> shift-invariant

- L266 left. Consider replacing *Embeddings’* with *Embedding*

- L380 right. No citation for ImageWoof dataset. (There's no paper for it so usually people cite the imagenette github repo.)

- L711 Need to adjust so the 1 and T in the sub/superscripts above one another don't run together. At the moment they look like one big, combined dagger symbol!


**References**

Be careful with casing (often messed up in the bibtex file and has to be manually corrected). Some names need to be title cased: Fréchet, Vendi, ImageNet. Some initialisms need to be changed from lower to upper case: GAN, PCA, COCO, BERT, CLIP. But the worst citation formatting is `Ro{bert}a`, which has the wrong casing and literal curly braces (also for `{bert}`)!

**Other Strengths And Weaknesses:**

**Strengths**

It was not initially obvious that this eigen-based clustering comparison would scale linearly with dataset size, but the proof and algorithm made it clear this was the case. This scaling is important for the utility of the method, since it is desirable to be able to deploy at scale.

**Weaknesses**

- The work on fine-tuning using SPEC seems a bit rushed within the paper, possibly it was a late addition and space was limited. I think the paper would benefit from further experiments exploring this component of the work. The fact that this can be done to induce clusterings from one encoder into another *without having to align their embedding spaces* is certainly a strength, but it is not highlighted in the paper.

- The paper should emphasize clearly that the distance metric is asymmetric.

- Fig 1 caption should show more detail about which classes and samples are selected to show in the tSNE plots.

**Questions For Authors:**

Why use this generated text data instead of an established dataset?

Why change from CLIP to OpenCLIP for Table 1?

**Relation To Broader Scientific Literature:**

The work is novel and relevant. This tool has the potential to be useful for the model interpretability community.

**Theoretical Claims:**

The proofs are in the appendix. I checked "A.2. Proof of Proposition 4.3", which I take to be correct.

Some steps in the main text eluded me and could be made clearer. For instance, L156 makes sense having read Proof of Proposition 4.3, but wouldn't otherwise. Also, it was unclear why a Gaussian RBF kernel needed to be used when other parts of the work were happy to support the kernel used being cosine-similarity.

---

> ### Author Rebuttal · Authors · 2025-04-01
>
> We thank Reviewer hTAA for the thoughtful and constructive feedback on our work. Below is our response to the comments and questions in the review: ([Our Numerical results are shown in this link](https://github.com/ICML6204/ICML6204/blob/main/ICML_Rebuttal.pdf))
>
> **1- Experiments on GPT-4o generated text data**
>
> We generated a dataset using GPT-4o, covering categories including professions, emotions, genders, actions, objects. We used the bellow prompt to generate. In the revised manuscript, we will explain the generation process.
>
> The prompt for generating the text dataset: “You are an expert in text-to-image models. Text-to-image models take a text prompt as input and generate images. Your task is to generate a prompt describing a person in [Profession] [Emotion], and [Gender] performing [Action] with [Object].
>
> **2- Additional experiment for real text datasets**
>
> To further address the reviewer’s comment, we also validated our approach on a large-scale real text dataset: WikiText-2. We split the dataset into 10K samples, each containing 100 tokens. Then, we used SPEC to compare CLIP and RoBERTa embeddings. The results can be found in Figure 2. We observed that RoBERTa better clustered Military Operations, Species Biology, Historical Figures, and Music, while CLIP embeddings more strongly clustered Sports and Science.
>
> We also examined the distribution of pairwise distances within each cluster to verify that one embedding successfully captured these clusters while the other was less inclined to do so. Also, we ran the K-means clustering algorithm 50 times on each of the embedding's features and computed the averaged (across the 50 runs) Normalized Mutual Information (NMI) between the K-means labels and the SPEC-identified labels. The results demonstrate that one embedding achieved considerably stronger alignment with KMeans labels.
>
> **3- Evaluation on SPEC-diff**
>
> To show that SPEC-diff effectively measures embedding similarity in SPEC-Align, we considered the SPEC-Align of CLIP to DINOv2 and tracked the SPEC-diff scores while plotting their kernel matrices. The results in Figure 3 show that as the kernel matrices look more similar, the SPEC-diff value decreases. The results suggest the SPEC-diff's utility for quantifying embedding alignment in SPEC-Align.
>
> **4- Explanation on the statement in Line 156**
>
> We will make the explanation clear that since the multiplication order in $C=\frac{1}{n}\Phi^\top \Phi$ and $C=\frac{1}{n}\Phi \Phi^\top$ is flipped, they share the same non-zero eigenvalues, and those eigenvectors are in one-to-one correspondence.
>
> **5- The choice of Gaussian vs. cosine similarity kernel functions**
>
> The choice of kernel function determines the similarity measure used for the clustering algorithm. The cosine similarity looks only at the angle between the vectors, while the Gaussian kernel concerns the Euclidean distance between the input points. The choice of the kernel function depends on the input embeddings and their induced geometry in the embedding space. As we discussed in the text, our analysis can be efficiently applied to both kernels.
>
> **6- Relationship to Laplacian spectral clustering**
>
> In the revised draft, we will make the connection to Laplacian spectral clustering more clear. One main difference between the proposed SPEC clustering and Laplacian spectral clustering is the usage of kernel matrix in SPEC (similar to Kernel PCA) vs. the Laplacian (kernel minus the diagonal degree matrix) in spectral clustering. We will explain this difference in the draft.
>
> **7- Additional results on embedding alignment using SPEC-Align**
>
> We aligned the CLIP text embedding with the T5-XL model. In Figure 6 of the rebuttal link (last page), we observe that the CLIP kernel has become more similar to T5-XL, and the SPEC-diff is also decreasing.
>
> **8- Symmetry of SPEC-diff**
>
> We would like to clarify that SPEC-diff is a symmetric measure because it is defined as the *spectral radius* (eigenvalue with maximum absolute value) of the difference of the two kernel matrices. Since the definition concerns the spectral radius (and not the maximum eigenvalue), the SPEC-diff is symmetric with respect to the embedding order. We will make this point clear in the writing.
>
> **9- Clarification on kernel values**
>
> We would like to clarify that the paper’s main results suppose a normalized kernel function where for every $x$, we have $k(x,x)= \langle \phi(x), \phi(x) \rangle=1$, i.e. $\Vert \phi(x) \Vert = 1$. Following Cauchy-Schwarz inequality, $| k(x,x’)|\le 1$ holds for every $x,x’\in\mathcal{X}$. We will make this point clear in the revised draft.
>
> **10- Typos and writing improvements**
>
> We thank the reviewer for pointing them out. We will correct them in the revised draft.
>
> **11-Change from CLIP to OpenCLIP**
>
> For alignment experiments, we utilized the widely-cited open source OpenCLIP github repository. We note that in Table 1, the CLIP backbones are the same.

---

> > ### Comment · Reviewer_hTAA · 2025-04-08
> >
> > I thank the authors for their response.
> >
> > 1-2. Thank you for adding the additional details on the GPT-4o prompt, and experiments on WikiText-2. This addresses my concern on this point. N.B. If the number of clusters which can be identified by SPEC and KMeans are not the same, I recommend using the Adjusted Mutual Information instead of NMI since the former corrects for the chance level agreement between clusterings.
> >
> > 3,4,6,7,9,10. Thank you for the clarifications and for adding this.
> >
> > 5. For sure, one should mention the relationship between the Gaussian RBF kernel and Euclidean norm in the text, to explain the use of a Gaussian beyond it merely being well known. I don't recall the specifics now, but it seemed that some of the theorems assumed a Gaussian kernel and others assumed cosine similarity and it wasn't clear why this was inconsistent. This was what I was referring to.
> >
> > 8. I think I was too brief and unclear before so to clarify. SPEC(A,B) involves taking the difference between the kernels for A and B and is asymmetric, resulting in graphs which are different for DINOv2-CLIP than for CLIP-DINOv2. The methodology doesn't find differences, it finds clusters that are prevalent in A but not B, thus SPEC(A,B) is different from SPEC(B,A). Consequently, SPEC-diff(A,B) is also not the same as SPEC-diff(B,A). This asymmetry with respect to their arguments means neither SPEC nor SPEC-diff are distance metrics, just like how KL-divergence is a useful measurement but isn't a distance metric. The authors don't describe either SPEC nor SPEC-diff as a distance metric in the paper, but they do refer to it as a **distance measure** several times, e.g. the paragraph starting L064 left. Hence I think it would be prudent to point out it at this point in the paper which says "distance measure" that SPEC is asymmetric and hence isn't a distance metric, to forestall potential confusion on this point.
> >
> > > 11-Change from CLIP to OpenCLIP
> > >
> > > For alignment experiments, we utilized the widely-cited open source OpenCLIP github repository. We note that in Table 1, the CLIP backbones are the same.
> >
> > 11. No, in Table 1 it says the OpenCLIP backbone is the same as the OpenCLIP backbone and the CLIP backbone is not mentioned. Having looked at it again, the methodology for this section (L425, Aligning embeddings using SPEC-align) is still very unclear. I really don't understand what experiment has been done and shown in Table 1. If I have to guess, I can only assume the reason for the switch to OpenCLIP for this experiment was you needed to know the training data and script to use, since that is publicly known for OpenCLIP but not CLIP. Otherwise the experiment could have been done with CLIP since it is an open-weight model. Did you train your own version of OpenCLIP from scratch with SPEC-align throughout? Or did you fine-tune OpenCLIP with SPEC-align? If it is a fine-tuning experiment, why couldn't you fine-tune the CLIP model for consistency with the rest of the paper and the first half of this section, and why does Table 1 only say "LAION 400M" for SPEC-align[ed] OpenCLIP, instead of both LAION 400M and MS-COCO 2017? The methodology provided in this paragraph (L412, right) is "we conducted an experiment similar to (Oquab et al., 2024)", a citation of DINOv2, which I assume is only intended indicate the evaluation methodology and not any of the training methodology. Is this "SPEC-aligned OpenCLIP" model shown in Table 1 the same as the model for which the "SPEC-align CLIP Kernel" is shown in Figure 3? If so, why do they have different names? If not, why does the first paragraph (L436, left) link to Appendix B.4.1, which exclusively discusses OpenCLIP.
> > The only thing that I'm sure on from this section is I think you need to add a citation to OpenCLIP to the reference list, either the [Zenodo for the repo](https://zenodo.org/badge/latestdoi/390536799), or their [scaling paper](https://openaccess.thecvf.com/content/CVPR2023/html/Cherti_Reproducible_Scaling_Laws_for_Contrastive_Language-Image_Learning_CVPR_2023_paper.html).
> >
> > 12. Additional point: t-SNE (2002) is overly sensitive to the choice of the perplexity parameter, which can result in misleading plots when trying to observe by eye the strength of clustering in plots. I recommend considering using [PaCMAP (2021)](https://arxiv.org/abs/2012.04456) instead, as this is a newer and more stable/reliable dimensionality reduction method which does a better job at retaining the structure of data when projecting it down to 2d, without needing as much fine-tuning of its parameters. I should have mentioned this before, but didn't want to nitpick unnecessarily.
> >
> > Other.
> > I mentioned before that the subscript was missing for $\Gamma\Psi\Psi$ at L190. Not that it is also incorrect at L231 right.
> >
> > As there are still snags which I feel should be addressed, I will keep my score for now.

---

> > > ### Author Response · Authors · 2025-04-09
> > >
> > > We sincerely thank Reviewer hTAA for the thoughtful feedback on our response. Regarding the points in the reviewer’s feedback:
> > >
> > > **Adjusted Mutual Information**
> > >
> > > We thank Reviewer hTAA for the suggestion. In the revision, we will report the Adjusted MI in the cases where the KMeans clustering is run to find fewer numbers of clusters compared to the number of SPEC-identified clusters.
> > >
> > > **The types of kernel functions in the theorems**
> > >
> > > We would like to clarify that only Proposition 4.3 exclusively holds for finite-dimension kernels, that does not apply to the Gaussian (RBF) kernel. Theorem 4.4 aims to address this gap by using the proxy (finite) random Fourier features that applies to any shift-invariant kernel such as the Gaussian kernel. Except for these results, the rest of our theoretical discussion applies to a general kernel function. We will make this point clear in the revised text.
> > >
> > > **Symmetry in SPEC(A,B) vs. SPEC-diff(A,B)**
> > >
> > > The reviewer is absolutely right that the kernel difference matrix in SPEC$(A,B) = K_A - K_B$ is not symmetric, and the eigenvalues can have a non-symmetric distribution around 0.
> > >
> > > On the other hand, regarding the (scalar) SPEC-diff measure, we indeed have SPEC-diff(A,B)=SPEC-diff(B,A). This is because SPEC-diff is the spectral radius of $K_A - K_B$, which is an even function equal to the spectral radius of $K_B - K_A$. However, since the spectral radius does not satisfy the triangle inequality, SPEC-diff does not provide a metric distance. In the revision, we will clarify that SPEC-diff is a symmetric pseudo-distance which does not satisfy the triangle inequality.
> > >
> > > **Experiments in Figure 3 and Table 1**
> > >
> > > We thank the reviewer for pointing this out. We would like to clarify that Figure 3’s alignment results of CLIP to DINOv2 has been performed on the ImageNet dataset (not the MS-COCO dataset used to align OpenCLIP [1] in Table 1). A better positioning of Figure 3 would be right after Figure 2, because the two figures share the same problem setting. We will revise and position Figure 3 right after Figure 2 to avoid any confusion.
> > >
> > > **Results on aligning OpenCLIP in Table 1**
> > >
> > > We appreciate the reviewer’s comment on the switch to the OpenCLIP model in Table 1. First, we would like to clarify that the experiment in Table 1 uses the **MS-COCO 2017 dataset** for the kernel-based alignment to DINOv2. The reason we reported the OpenCLIP results in Table 1 is the considerable gain in the linear-probe ImageNet accuracy results for the aligned OpenCLIP. Although the fine-tuning of OpenCLIP has been performed on MS-COCO, the linear-probe ImageNet accuracy gain was significant as we reported in Table 1.
> > >
> > > On the other hand, in our experiments, the same MS-COCO-dataset fine-tuning of the (OpenAI) CLIP model did not lead to a significant gain in ImageNet accuracy. The original CLIP ImageNet linear-probe accuracy was 67.2%, which only changed to 67.4% accuracy in the fine-tuning on the MS-COCO dataset. Still, we would like to highlight that the cluster similarity between the aligned (OpenAI) CLIP and DINOv2 boosted significantly in this experiment, as the SPEC-diff value reduced from 0.49 to 0.03 during the alignment. However, for the OpenCLIP case, both the SPEC-diff and ImageNet accuracy simultaneously improved, as reported in the text.
> > >
> > > Regarding these numerical observations, we hypothesise that the different ImageNet accuracy of CLIP vs. OpenCLIP is due to the (unknown) training data of CLIP and the distribution mismatch between ImageNet (testing data) and MS-COCO (fine-tuning data). To validate this hypothesis, we used the SPEC-Align on the ImageNet data (instead of the MS-COCO 2017 dataset) for aligning CLIP to DINOv2, and we observed that the ImageNet linear-probe accuracy could jump from 67.2% to 73.9% after only 4 epochs of the alignment fine-tuning.
> > >
> > > We will include the above discussion in the revised paper. Also, we will only discuss the (OpenAI) CLIP model results in the revised main text, and will defer the numerical results of OpenCLIP in Table 1 to the Appendix to ensure the main text’s experimental results have sufficient consistency.
> > >
> > > **Reference of OpenCLIP model**
> > >
> > > We will include the reference paper and GitHub repository and the paper of OpenCLIP [1] in the revised text.
> > >
> > > **Using PaCMAP for dimensionality reduction and visualization of clusters**
> > >
> > > We thank Reviewer hTAA for the suggestion. We will include the PaCMAP visualizations in the revised text.
> > >
> > > **Missing subscript**
> > >
> > > We thank the reviewer for catching the missing subscript. We will fix it in the revision.
> > >
> > > [1] Cherti et al., “Reproducible Scaling Laws for Contrastive Language-Image Learning” CVPR 2023

---

### Decision · Program_Chairs · 2025-05-01

**Decision:**

Accept (poster)

**Comment:**

The paper proposes a spectral framework to compare and contrast feature embeddings for interpretable ML.  The methodology is novel and has wide potential applications.  The general concensus of the reviewers is that this submission adds to the growing and important literature on interpretable machine learning.